# PBN-PVT projections modulate negative affective states in mice

**Ya-Bing Zhu[1†], Yan Wang[1†], Xiao-Xiao Hua[2†], Ling Xu[1], Ming-Zhe Liu[3], Rui Zhang[1], Peng-Fei Liu[1], Jin-Bao Li[1], Ling Zhang[2]\*, Di Mu[4,5]\***

[1]Department of Anesthesiology, Shanghai General Hospital, Shanghai Jiao Tong University School of Medicine, Shanghai, China; [2]The First Rehabilitation Hospital of Shanghai, Tongji University School of Medicine, Shanghai, China; [3]Department of Respiratory, The First Affiliated Hospital of Guangzhou Medical University, Guangzhou, China; [4]SUSTech Center for Pain Medicine, School of Medicine, Southern University of Science and Technology, Shenzhen, China; [5]Department of Anesthesiology, Shanghai General Hospital, Shanghai Jiao Tong University School of Medicine, Shanghai, China

**\*For correspondence:**
lzhang0808@tongji.edu.cn (LZ);
damonmu@163.com (DM)

[†]These authors contributed equally to this work

**Competing interest:** The authors declare that no competing interests exist.

**Abstract** Long-lasting negative affections dampen enthusiasm for life, and dealing with negative affective states is essential for individual survival. The parabrachial nucleus (PBN) and thalamic paraventricular nucleus (PVT) are critical for modulating affective states in mice. However, the functional roles of PBN-PVT projections in modulating affective states remain elusive. Here, we show that PBN neurons send dense projection fibers to the PVT and form direct excitatory synapses with PVT neurons. Activation of the PBN-PVT pathway induces robust behaviors associated with negative affective states without affecting nociceptive behaviors. Inhibition of the PBN-PVT pathway reduces aversion-like and fear-like behaviors. Furthermore, the PVT neurons innervated by the PBN are activated by aversive stimulation, and activation of PBN-PVT projections enhances the neuronal activity of PVT neurons in response to the aversive stimulus. Consistently, activation of PVT neurons that received PBN-PVT projections induces anxiety-like behaviors. Thus, our study indicates that PBN-PVT projections modulate negative affective states in mice.

## Editor's evaluation

This study will interest neuroscientists, in particular those interested in the neural circuits that support emotional processing. Using modern neuroscience techniques, the authors demonstrate that anatomical projections from the parabrachial nucleus to the paraventricular nucleus thalamus contribute to aversive states like fear and anxiety. Overall, the study offers important details of a previously uncharacterized brain circuit.

## Introduction

Threat and injury often induce defensive behaviors, such as flight, freezing, hiding (**Ohman and Mineka, 2001**), and negative affective states, such as fear and anxiety (**Jimenez et al., 2018**). Such behavioral adaptations and psychological responses are essential for animal survival, and understanding their mechanisms is of great interest. It is worth noting that the parabrachial nucleus (PBN) in the brainstem plays a critical role in encoding danger signals and promoting affective behavior states to limit harm in response to potential threats (**Campos et al., 2018**).

The PBN receives the majority of the ascending inputs from the spinal cord (**Todd, 2010**), and PBN neurons respond robustly to nociception, food neophobia, hypercapnia, and threat to maintain

homeostasis in stressful circumstances (*Campos et al., 2018*; *Kaur et al., 2013*). The PBN relays this information (visceral malaise, taste, temperature, pain, itch) to brain areas, such as the hypothalamus, central amygdala (CeA), thalamus, insular cortex (IC), and periaqueductal gray (PAG), to participate in diverse physiological processes (*Chiang et al., 2019*; *Palmiter, 2018*; *Saper, 2016*). A recent study found that subpopulations of the PBN have distinct projection patterns and functions (*Chiang et al., 2020*). Neurons in the dorsal division of the PBN projecting to the ventromedial hypothalamus (VMH) and PAG mediate escape behaviors. In contrast, neurons in the external lateral division of the PBN projecting to the bed nucleus of the stria terminalis (BNST) and the CeA mediate aversion and avoidance memory (*Chiang et al., 2020*). Optogenetic manipulation of specific outputs from PBN accomplishes distinct functions (*Bowen et al., 2020*). In the thalamus, the intralaminar thalamus nucleus (ILN) is the downstream target of PBN neurons that receive spinal cord inputs, and this ILN pathway participates in nociception processing (*Deng et al., 2020*). In addition to the ILN, the thalamic paraventricular nucleus (PVT) is another primary target of projections from the PBN nucleus in the thalamus (*Chiang et al., 2020*).

The PVT has been implicated in a range of affective behaviors (*Hsu et al., 2014*). The functional roles of the PVT include modulation of a diverse array of processes, such as arousal (*Ren et al., 2018*), drug addiction (*Zhu et al., 2016*), reward seeking (*Do-Monte et al., 2017*; *Engelke et al., 2021*), stress (*Beas et al., 2018*; *Gao et al., 2020*), and associative learning and memory retrieval (*Penzo et al., 2015*; *Do-Monte et al., 2015*; *Zhu et al., 2018*; *Keyes et al., 2020*). The PVT receives a significant amount of input from the brainstem, hypothalamus, and prefrontal cortical areas and projects to the infralimbic cortex, nucleus accumbens (NAc), BNST, and CeA (*Kirouac, 2015*; *Vertes et al., 2015*). The convergent signals include arousal from the hypothalamus (*Ren et al., 2018*), emotional saliency from the prefrontal cortex (*Yamamuro et al., 2020*), and stress responsivity from the locus coeruleus (LC) (*Beas et al., 2018*), which might help promote appropriate behavioral responses to environmental challenges. However, despite substantial improvements in our understanding of the neurocircuitry of the PVT, the functional role of PBN-PVT projections remains mostly unknown.

In this study, we used viral tracing and electrophysiology to dissect the anatomical and functional connections between the PBN and the PVT. By using optogenetic and pharmacogenetic approaches, we then demonstrated that PBN-PVT projections modulate negative affective states in mice.

## Results

### Functional connectivity pattern of PBN-PVT projections

Although it has been shown that the PVT receives input from the PBN (*Chiang et al., 2020*; *Li and Kirouac, 2012*), the detailed morphology of PBN-PVT projections and whether these two nuclei form direct functional synapses remain unknown. To examine the synaptic connectivity between the PBN and the PVT, we injected AAV2/8-hSyn-ChR2-mCherry virus into the PBN and employed whole-cell patch-clamp recording. There were dense projection fibers in the PVT (*Figure 1A–C*) and precisely time-locked action potentials induced by brief laser pulses in PBN ChR2$^+$ neurons (5 Hz, 10 Hz, and 20 Hz; *Figure 1G*). We found that optogenetic activation of PBN projection fibers evoked excitatory postsynaptic currents (EPSCs) in 34 of 52 PVT neurons. The medial PVT (bregma: –0.94 to –1.82 mm; 18 of 25 cells, 72.0%) and posterior PVT (pPVT; bregma: –1.82 to –2.3 mm; 14 of 21 neurons, 66.7%) showed higher connectivity than the anterior PVT (bregma: –0.22 to –0.94 mm; 2 of 6 cells, 33.3%; *Figure 1D–F*). The average amplitude of the light-evoked EPSCs was 103.4 ± 11.93 pA (*Figure 1H*). Moreover, the latency of EPSCs was short with small jitter (*Figure 1I and J*), indicating monosynaptic connections between the PBN and PVT. Consistently, the EPSCs were sensitive to the Na$^+$ channel blocker tetrodotoxin (TTX, 1 µM) and were rescued by the K$^+$ channel blocker 4-aminopyridine (4-AP, 100 µM). The EPSCs were further blocked by the AMPA receptor antagonist NBQX (10 µM), confirming the monosynaptic glutamatergic innervation of PVT neurons by PBN neurons (*Figure 1K and L*). In addition, we also observed light-evoked inhibitory postsynaptic currents (IPSCs) in only 4 of 52 PVT neurons (less than 30 pA). Thus, these results indicate that most monosynaptic connections between PBN and PVT are glutamatergic.

Next, we asked what are the distribution pattern and molecular identity of PBN-PVT neurons. We injected retroAAV2/2-hSyn-Cre virus into the PVT of *Rosa26-tdTomato* mice, which could retrogradely label projection neurons in the PBN (*Figure 1M–P*). We found that tdTomato$^+$ neurons

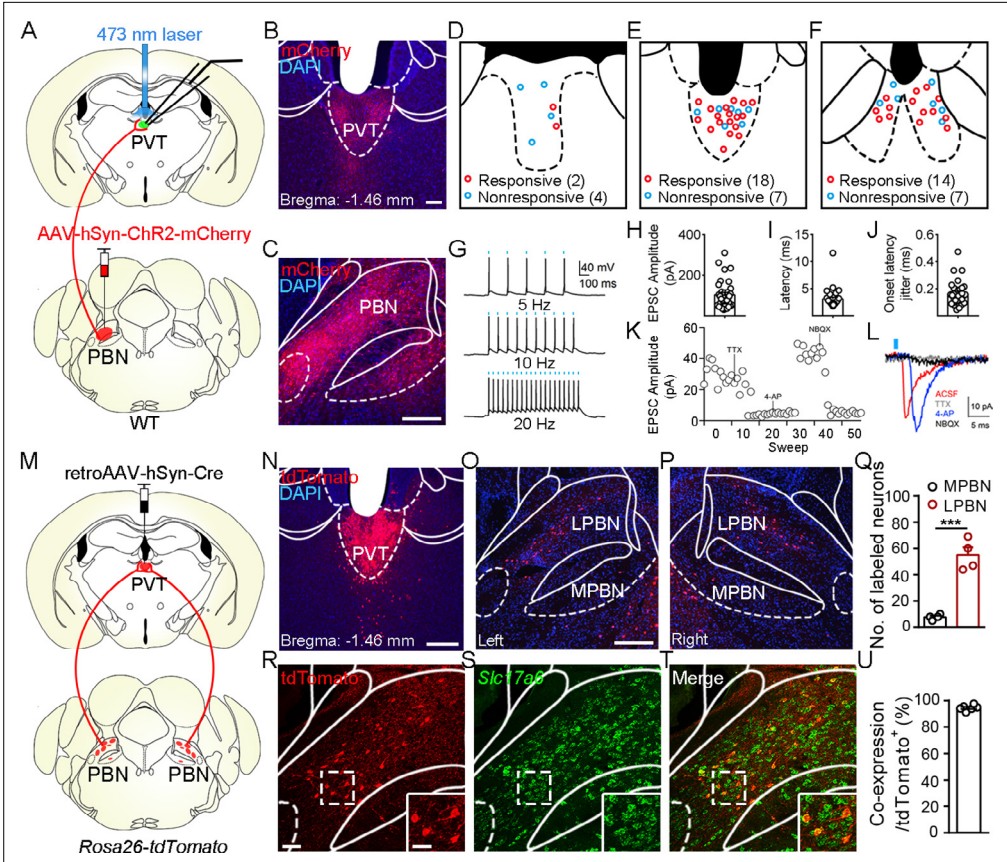

**Figure 1.** Functional connectivity pattern of parabrachial nucleus-paraventricular nucleus (PBN-PVT) projections.
(**A**) The schematic for virus injection of AAV2/8-hSyn-ChR2-mCherry into the PBN nucleus and the slice recording with 473 nm laser stimulation. (**B**) The projection fibers in the PVT nucleus. Scale bar: 100 µm. (**C**) The AAV2/8-hSyn-ChR2-mCherry virus expression in the PBN nucleus. Scale bar: 200 µm. (**D−F**) The locations of the recorded cells in the anterior PVT (**D**), the middle PVT (**E**), and the posterior PVT (**F**). Red circles indicate neurons with excitatory postsynaptic currents (EPSCs), and blue circles indicate neurons without EPSCs. (**G**) The 473 nm laser-induced time-locked action potential firing at 5 Hz (top), 10 Hz (middle), and 20 Hz (bottom) in the ChR2-expressing neuron in the PBN. Scale bars: 100 ms, 40 mV. (**H−J**) The amplitude of light-evoked EPSCs (**H**), the latency of EPSCs (**I**), and the latency jitter of EPSCs (**J**) from all 34 responsive neurons in the PVT. (**K**) Amplitudes of light-evoked EPSCs recorded from a PVT neuron (right panel). (**L**) The light-evoked EPSC was completely blocked by 1 µM tetrodotoxin (TTX), rescued by 100 µM 4-aminopyridine (4-AP), and blocked by 10 µM NBQX (AMPA/ kainate receptor antagonist). Scale bars: 5 ms, 10 pA. (**M**) Schematic shows retroAAV2/2-hSyn-Cre injection into the PVT nucleus on *Rosa26-tdTomato* mice. (**N**) The injection site in the PVT nucleus. Scale bar: 200 µm. (**O, P**) The distribution of the tdTomato-positive neurons in the left PBN (**O**) and the right PBN (**P**). (**Q**) The quantification of the tdTomato-positive neurons in the lateral PBN (LPBN) and the media PBN (MPBN). n = 4 mice. Scale bar: 200 µm. (**R−T**) Double staining of tdTomato with *Slc17a6* mRNA by in situ hybridization. Scale bar: 50 µm, the scale bar in the quadrangle was 25 µm. (**U**) Quantification of the double-positive neurons over the total number of tdTomato-positive neurons, n = 6 sections from three mice. ***p<0.001, data are represented as mean ± SEM. Paired Student's *t*-test for (**Q**).

The online version of this article includes the following figure supplement(s) for figure 1:

**Figure supplement 1.** Characterization of parabrachial nucleus-paraventricular nucleus (PBN-PVT) neurons.

**Figure supplement 2.** The distribution pattern of the parabrachial nucleus-paraventricular nucleus (PBN-PVT) glutamatergic projection.

**Figure supplement 3.** The distribution pattern of collateral projection fibers from parabrachial nucleus-paraventricular nucleus (PBN-PVT) neurons.

were bilaterally located in the lateral PBN (55 ± 6 neurons, n = 4 mice) and rarely in the medial PBN (8 ± 1 neurons; *Figure 1O–Q*). These results indicate that most PVT inputs originate in the lateral aspect of the PBN. We then performed tdTomato staining and in situ hybridization with *Slc17a6* (Vglut2) probe and found that approximately 94.4% of tdTomato+ neurons expressed *Slc17a6* mRNA (Vglut2+, *Figure 1R–U*). We also examined several markers for subpopulations of PBN neurons, including tachykinin 1 receptor (Tacr1), tachykinin 1 (Tac1), prodynorphin (Pdyn), and calcitonin gene-related peptide (CGRP). We found that tdTomato+ neurons were only partially co-labeled with *Tacr1* mRNA (38.0%), *Tac1* mRNA (6.4%), or *Pdyn* mRNA (23.0%) and not with CGRP protein (*Figure 1—figure supplement 1*). These results indicate that the majority of PBN-PVT neurons are glutamatergic neurons, and most of these neurons are not labeled by Tacr1, Tac1, Pdyn, or CGRP.

To examine the collateral projections from the PVT-projecting PBN glutamatergic neurons, we injected AAV2/8-EF1α-DIO-EGFP virus into the PBN of *Slc17a6^tm2(cre)Lowl* (also called Vglut2-ires-Cre) mice (*Figure 1—figure supplement 2A*). Robust expression of AAV2/8-EF1α-DIO-EGFP was found in both the lateral and medial PBN (*Figure 1—figure supplement 2B–D*). It is worth noting that the density of EGFP+ fibers was higher in the middle and pPVT (*Figure 1—figure supplement 2E–H*), considering the notion that the pPVT is particularly sensitive to aversion (*Gao et al., 2020*). We also found collateral projections from PBN-PVT neurons in the BNST, lateral hypothalamus (LH), paraventricular nucleus of the hypothalamus (PVN), and PAG but not in the CeA or VMH (*Figure 1—figure supplement 3*).

## Optogenetic activation of PBN-PVT projections induces anxiety-like behaviors and aversion-like behaviors

To examine the functional role of PBN-PVT projections in modulating affective behaviors, we injected AAV2/9-EF1α-DIO-ChR2-mCherry virus or AAV2/9-EF1α-DIO-mCherry virus bilaterally into the PBN of Vglut2-ires-Cre mice and implanted optic fibers above the PVT to selectively activate PBN-PVT projections (*Figure 2A*). Four weeks after surgery, we found robust expression of ChR2-mCherry (*Figure 2B and C*, *Figure 2—figure supplement 1A*) or mCherry (*Figure 2—figure supplement 1C*) in bilateral PBN neurons and axon terminals in the PVT (*Figure 2D*, *Figure 2—figure supplement 1B and D*).

We performed a 15 min optogenetic manipulation in the open field test (OFT; 0–5 min laser off, 5–10 min laser on, 10–15 min laser off; *Figure 2E*). Optogenetic activation (473 nm, 20 Hz, 5 mW, 5 ms) of the efferents from the PBN to the PVT elicited instant running behavior along the chamber wall with a significantly increased velocity (*Figure 2F–H*, *Figure 2—video 1*). The activation of PBN-PVT projections also reduced the center time compared with that of the control mice (*Figure 2I*). It is worth noting that the velocity returned to normal once the laser was off, but the time spent in the center was still lower than that of the control group during the 5 min after stimulation. These results indicate that anxiety could last for at least several minutes after acute activation of PBN-PVT projections. Although the speed increased during the laser on period, the immobility time of the ChR2 mice during the laser on period was also increased (*Figure 2—figure supplement 2A*). Therefore, the distance during the laser on period and the total distance in 15 min were not changed (*Figure 2—figure supplement 2B*).

To provide a more detailed profile of behavior in the OFT, we further divided the laser on period (5–10 min) into five 1-min periods and analyzed the velocity, immobility time, center time, distance, and jumping (*Figure 2—figure supplement 2C–G*). We found that the velocity and immobility time were increased, and the center time was decreased in the ChR2 mice during most periods (*Figure 2—figure supplement 2C–E*). Furthermore, we observed that the distance and jumping behaviors were increased mainly in the first 1-min period in ChR2 mice (*Figure 2—figure supplement 2F and G*). This detailed analysis indicates that optogenetic activation of PBN-PVT projections could induce brief and robust running, jumping behaviors, and persistent anxiety-like behaviors, such as less time spent in the center.

In addition to anxiety, another critical component of negative affective states is aversion. Therefore, we used the real-time place aversion (RTPA) test to explore the function of optogenetic activation of PBN-PVT projections in modulating aversion (*Figure 2J*). We found that the activation of PBN-PVT projections reduced the time spent in the laser-paired chamber, and the aversion disappeared when the laser was off (*Figure 2K–M*, *Figure 2—video 2*). We also used a prolonged conditioning protocol that mimics drug-induced conditioned place aversion (CPA; *Figure 2—figure supplement 2H*). We

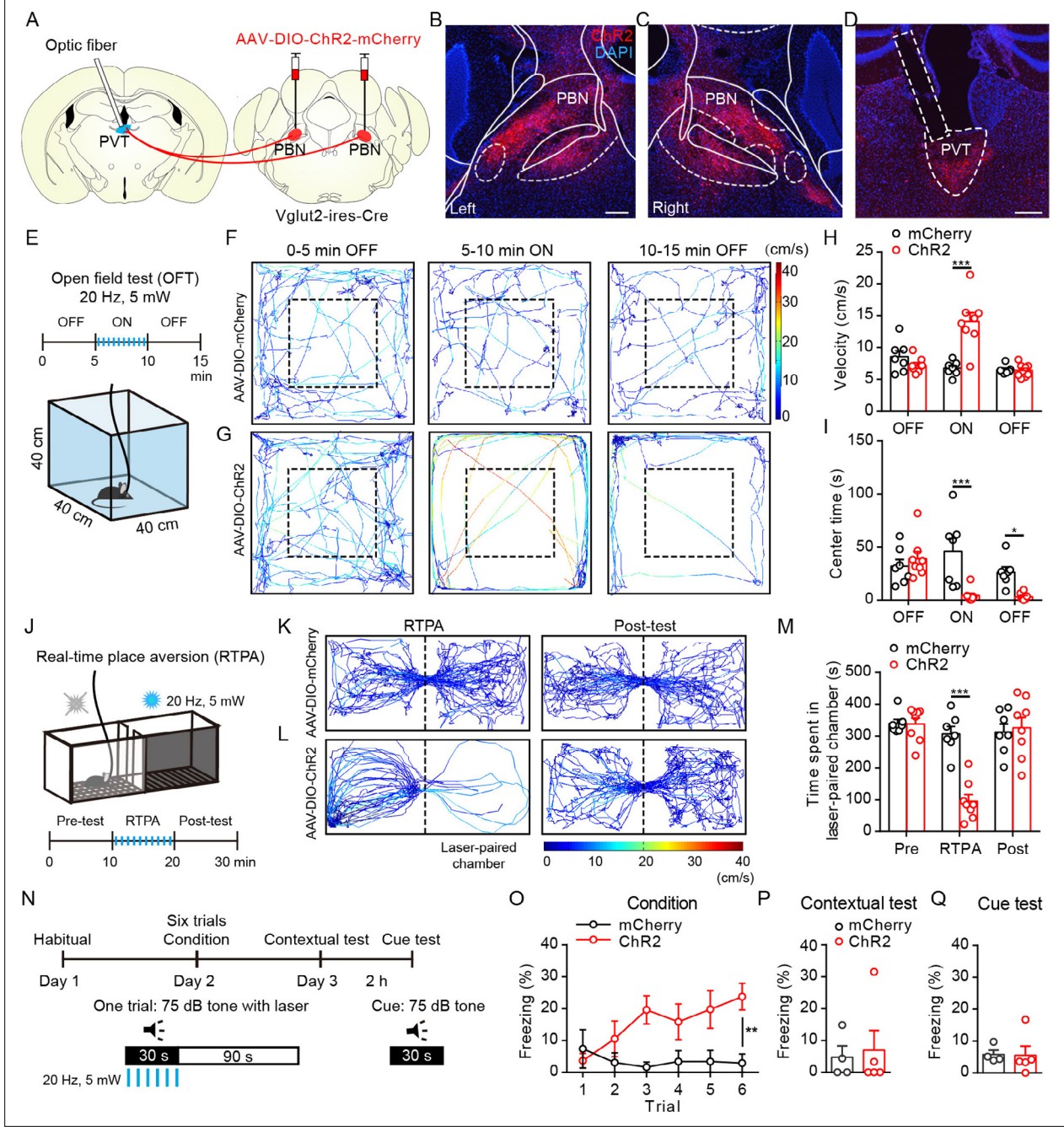

**Figure 2.** Optogenetic activation of parabrachial nucleus-paraventricular nucleus (PBN-PVT) projections induced negative affective states. (**A**) The illustration shows the injection of AAV2/9-EF1α-DIO-ChR2-mCherry virus into the PBN nucleus and the optic fiber above the PVT on the Vglut2-ires-Cre mice. (**B, C**) The virus injection sites of the left PBN (**B**) and the right PBN (**C**). Scale bar: 200 μm. (**D**) The projection axons from the PBN and the location of the optic fiber (rectangle) in the PVT. Scale bar: 200 μm. (**E**) The schematic of the open field test (OFT) with optogenetic activation via a 473 nm laser (20 Hz, 5 mW, 5 ms). (**F, G**) The example traces of the 15 min optogenetic manipulation OFT from an AAV2/9-EF1α-DIO-mCherry virus-injected mouse (**F**) or an AAV2/9-EF1α-DIO-ChR2-mCherry virus-injected mouse (**G**). (**H, I**) Quantification of the velocity (**H**) and the center time (**I**) in the OFT, mCherry group: n = 7 mice; ChR2 group: n = 8 mice. (**J**) The illustration of the real-time place aversion (RTPA) test with optogenetic activation via a 473 nm laser. The right side was paired with the laser. (**K, L**) The example traces of the RTPA and post-test from the mice injected with AAV2/9-EF1α-DIO-mCherry (**K**) or AAV2/9-EF1α-DIO-ChR2-mCherry (**L**). (**M**) Quantification of the time spent in the laser-paired chamber in the pre-test (Pre), RTPA, and post-test (Post), mCherry group: n = 7 mice; ChR2 group: n = 8 mice. (**N**) Schematic timeline of cue-dependent optogenetic conditioning. (**O**) Conditioned-freezing responses to sound cue paired with optogenetic activation of the PBN-PVT projection during training, mCherry group: n = 4 mice; ChR2 group: n = 5 mice. (**P, Q**) Optogenetic activation of the projection fibers from the PBN in the PVT did not induce context-dependent fear (**P**) and

*Figure 2 continued on next page*

*Figure 2 continued*

cue-dependent fear (**Q**), mCherry group: n = 4 mice; ChR2 group: n = 5 mice. *p<0.05, **p<0.01, ***p<0.001, all data are represented as mean ± SEM. Two-way ANOVA followed by Bonferroni test for (**H**), (**I**), (**M**), and (**O**). Unpaired Student's *t*-test for (**P**) and (**Q**).

The online version of this article includes the following video and figure supplement(s) for figure 2:

**Figure supplement 1.** Virus expression in the parabrachial nucleus (PBN) and the optic fiber position in the paraventricular nucleus (PVT) of Vglut2-ires-Cre mice injected with AAV2/9-EF1α-DIO-ChR2-mCherry virus or AAV2/9-EF1α-DIO-mCherry virus.

**Figure supplement 2.** Effects of optogenetic activation of parabrachial nucleus-paraventricular nucleus (PBN-PVT) projections in the open field test (OFT) and the conditioned place aversion (CPA).

**Figure 2—video 1.** Optogenetic activation of parabrachial nucleus-paraventricular nucleus (PBN-PVT) projections in open field test (OFT).
https://elifesciences.org/articles/68372/figures#fig2video1

**Figure 2—video 2.** Optogenetic activation of parabrachial nucleus-paraventricular nucleus (PBN-PVT) projections in real-time place aversion (RTPA).
https://elifesciences.org/articles/68372/figures#fig2video2

found that prolonged activation of PBN-PVT projections did not display aversion in the postconditioning test (*Figure 2—figure supplement 2I*).

To further confirm this instant aversion phenomenon, we subjected mice to the cue-dependent optogenetic conditioning test (*Figure 2N*). A 30 s auditory conditioning stimulus (CS) co-terminated with 30 s of synchronous optogenetic activation of PBN-PVT projections (laser stimulus [LS]) in this test. Activation of the PBN-PVT projections induced significant freezing behavior during six CS-LS pairings (*Figure 2O*). However, the freezing behavior to the same context or to the auditory cue in a novel context disappeared on the second day (*Figure 2P and Q*). These results indicate that optogenetic activation of PBN-PVT projections induces instant aversion and freezing but does not drive associative fear memory formation.

## Pharmacogenetic activation of PBN-PVT neurons induces anxiety-like behaviors and freezing behaviors

To further confirm the effects of activating the PBN-PVT pathway, we also used retrograde viral tracing and pharmacogenetic manipulation. We first injected retroAAV2/2-hSyn-Cre virus into the PVT and AAV2/9-hSyn-DIO-hM3Dq-mCherry virus or control virus bilaterally into PBN to specifically transduce PBN-PVT neurons with the hM3Dq, a designer receptor exclusively activated by designer drugs (*Figure 3A*; *Armbruster et al., 2007*). PBN-PVT neurons could be activated by intraperitoneal (i.p.) injection of clozapine N-oxide dihydrochloride (CNO; *Figure 3B–D*). The region of virus expression in the PBN is shown in *Figure 3—figure supplement 1*. Consistent with the optogenetic activation results, pharmacogenetic activation of PBN-PVT neurons reduced the center time, increased immobility time, and reduced the travel distance in the OFT (*Figure 3E–I*). At the same time, the velocities were not significantly different (*Figure 3J*). We also found that activation of PBN-PVT neurons did not affect motor ability in the rotarod test (*Figure 3—figure supplement 2G*). In addition, pharmacogenetic activation of PBN-PVT neurons decreased exploration time of open quadrants in the elevated zero maze (EZM; *Figure 3K and L*), further suggesting that activation of PBN-PVT neurons induces anxiety-like behaviors.

We further evaluated freezing behaviors in the fear conditioning chamber and found that pharmacogenetic activation of PBN-PVT neurons induced more freezing behaviors (*Figure 3M and N*). Although activation of PBN-PVT neurons induced significant anxiety-like behavior, it did not affect the depressive-like behaviors evaluated by the tail suspension test (TST; *Figure 3—figure supplement 2A*) or the forced swimming test (FST; *Figure 3—figure supplement 2B*). Previous studies have revealed that the PBN receives direct projections from the spinal cord and plays a vital role in pain processing (*Deng et al., 2020*; *Sun et al., 2020*). We then assessed whether pharmacogenetic activation of PBN-PVT neurons affected nociceptive behaviors. By performing the von Frey test and Hargreaves test, we found that the basal nociceptive thresholds were not affected after pharmacogenetic activation of PBN-PVT neurons (*Figure 3—figure supplement 2C and D*). Given the distinct mechanisms between the reflexive and coping responses induced by nociceptive stimulation (*Huang et al., 2019*), we injected formalin into the paw to induce inflammatory pain. We found that activation of PBN-PVT neurons did not affect formalin-evoked licking behaviors (*Figure 3—figure supplement 2E and F*).

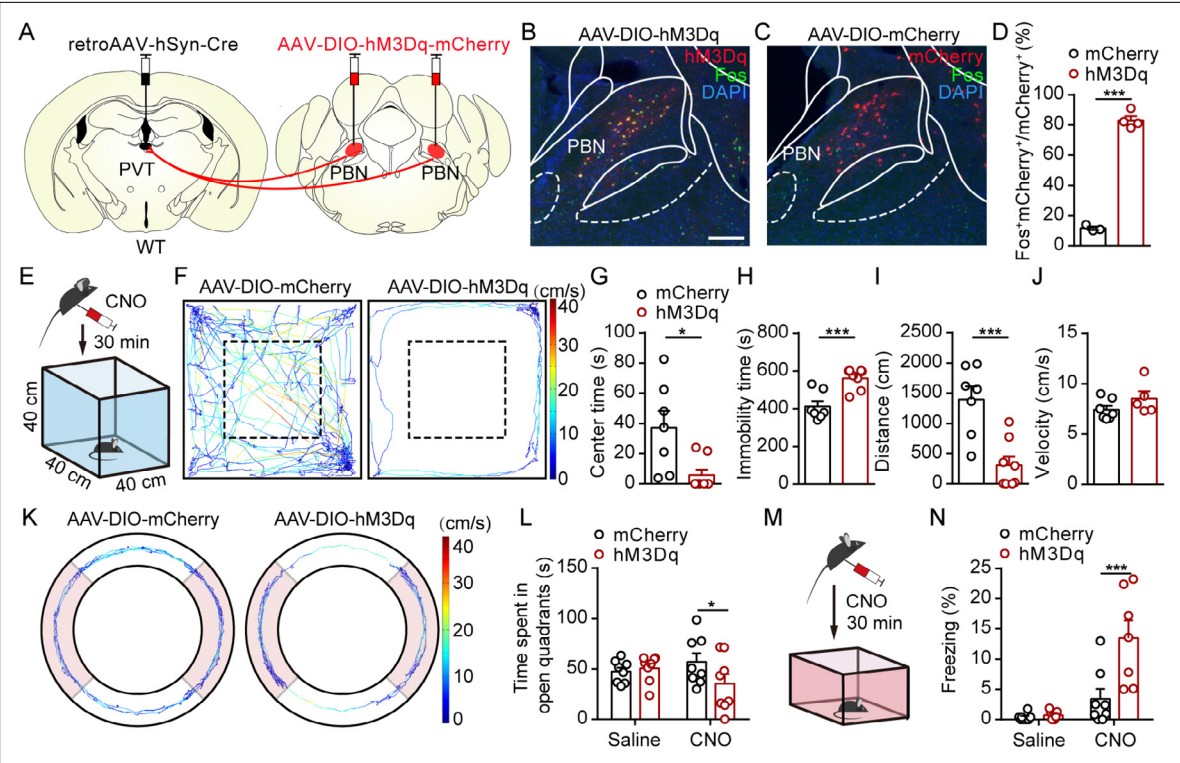

**Figure 3.** Pharmacogenetic activation of parabrachial nucleus-paraventricular nucleus (PBN-PVT) neurons induced anxiety-like behaviors and fear-like behaviors. (**A**) The illustration shows virus injection of retroAAV2/2-hSyn-Cre into the PVT nucleus and bilateral injection of AAV2/9-hSyn-DIO-hM3Dq-mCherry into the PBN nucleus. (**B, C**) Clozapine N-oxide dihydrochloride (CNO) administration evokes Fos expression in AAV2/9-hSyn-DIO-hM3Dq-mCherry-injected mice (**B**) but not in AAV2/9-EF1α-DIO-mCherry-injected mice (**C**). Scale bar: 200 µm. (**D**) Percentage of co-labeled neurons in the PBN, mCherry group: n = 3 mice; hM3Dq group: n = 4 mice. (**E**) The illustration of the open field test (OFT) with pharmacogenetic activation. (**F**) Example of the OFT traces from the mice infected with AAV2/9-EF1α-DIO-mCherry or AAV2/9-hSyn-DIO-hM3Dq-mCherry. (**G–I**) Quantification of the center time (**G**), the immobility time (**H**), and the total distance (**I**) in the OFT, mCherry group: n = 7 mice; hM3Dq group: n = 8 mice. (**J**) Quantification of the velocity in the OFT, mCherry group: n = 7 mice; hM3Dq group: n = 5 mice. (**K**) Example elevated zero maze (EZM) traces from the mice infected with AAV2/9-EF1α-DIO-mCherry and AAV2/9-hSyn-DIO-hM3Dq-mCherry. (**L**) Quantification of the time spent in open quadrants in the EZM test, n = 8 mice per group. (**M**) The illustration of pharmacogenetic activation-induced fear-like freezing behavior. (**N**) Pharmacogenetic activation of PBN-PVT neurons induced fear-like freezing behaviors, mCherry group: n = 8 mice; hM3Dq group: n = 7 mice. *p<0.05, ***p<0.001, all data are represented as mean ± SEM. Unpaired Student's *t*-test for (**D**), (**G**), (**H**), (**I**), and (**J**). Two-way ANOVA followed by Bonferroni test for (**L**) and (**N**).

The online version of this article includes the following figure supplement(s) for figure 3:

**Figure supplement 1.** Virus expression in the parabrachial nucleus (PBN) of mice injected with AAV2/9-hSyn-DIO-hM3Dq-mCherry or AAV2/9-EF1α-DIO-mCherry in the pharmacogenetic manipulation experiment.

**Figure supplement 2.** Pharmacogenetic activation of parabrachial nucleus-paraventricular nucleus (PBN-PVT) neurons did not affect depressive-like behaviors, basal nociceptive thresholds, formalin-induced licking behavior, or motor function.

These results indicate that activating the PBN-PVT pathway induces anxiety-like and freezing behaviors but not nociceptive behaviors.

## Inhibition of PBN-PVT projections reduces aversion-like behaviors and freezing behaviors

We next asked whether inhibition of PBN-PVT projections could modulate negative affective states. We first injected AAV2/9-EF1α-DIO-NpHR3.0-EYFP virus or AAV2/8-EF1α-DIO-EGFP virus bilaterally into the PBN and implanted optic fibers into the PVT of Vglut2-ires-Cre mice (*Figure 4A–C*, *Figure 4—figure supplement 1*). We used 2-methyl-2-thiazoline (2-MT), a widely used odorant molecule that can generate innate fear-like freezing responses in rodents (*Isosaka et al., 2015*), to induce a fear-like state. We found that 589 nm laser-induced inhibition of PBN-PVT projections reduced the aversion caused by 2-MT exposure (*Figure 4D–F*) and increased the moving duration (*Figure 4G*). We also observed that inhibition of PBN-PVT projections increased the time spent in the 2-MT zone in the OFT

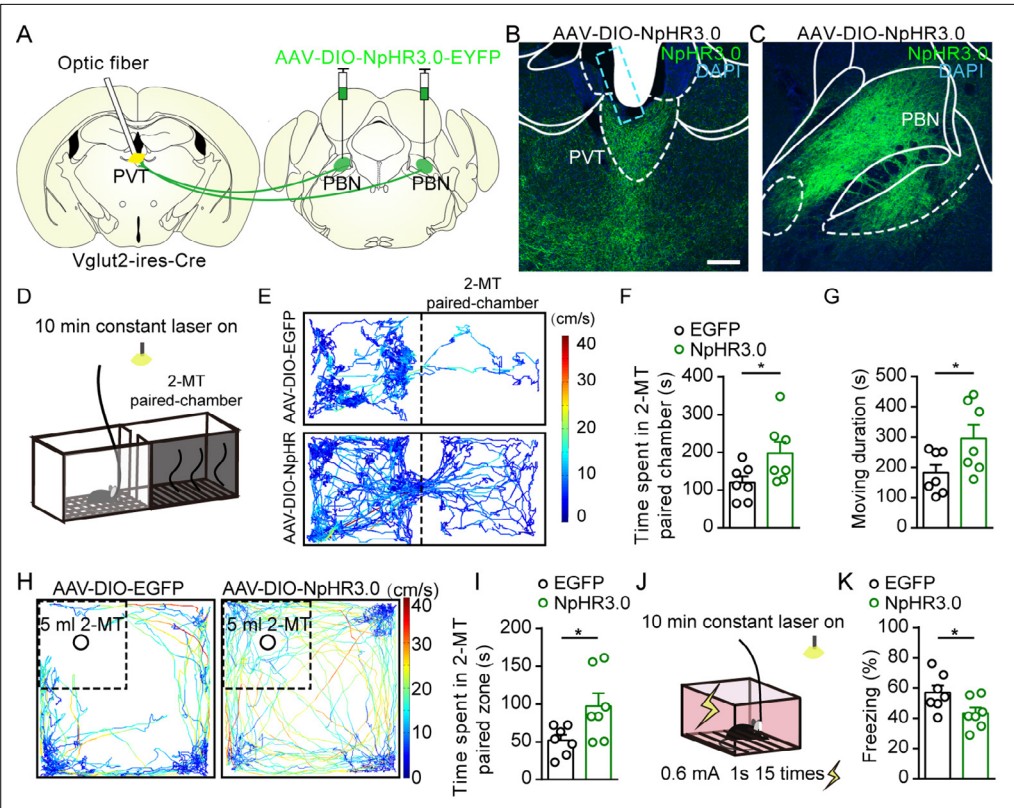

**Figure 4.** Optogenetic inhibition of parabrachial nucleus-paraventricular nucleus (PBN-PVT) projections reduced aversion-like behavior and fear-like behaviors. (**A**) The illustration shows the bilateral injection of AAV2/9-EF1α-DIO-NpHR3.0-EYFP virus into the PBN and placement of optic fiber above the PVT on Vglut2-ires-Cre mice. (**B, C**) Examples of AAV2/9-EF1α-DIO-NpHR3.0-EYFP expression in the PVT (**B**) and PBN (**C**). The cyan rectangle represents the position of the optic fiber. Scale bar: 200 μm. (**D**) Schematic of 2-methyl-2-thiazoline (2-MT)-induced aversion test with optogenetic inhibition via the 589 nm laser. (**E**) Representative traces of the mice infected with AAV2/8-EF1α-DIO-EGFP or AAV2/9-EF1α-DIO-NpHR3.0-EYFP in two chambers. (**F, G**) Quantification of the time spent in the 2-MT paired chamber (**F**) and the total moving duration (**G**), n = 7 mice per group. *p<0.05, **p<0.01, (**H**) Representative traces of the mice infected with AAV2/8-EF1α-DIO-EGFP or AAV2/9-EF1α-DIO-NpHR3.0-EYFP in the open field test (OFT) chamber. (**I**) Quantification of the time spent in the 2-MT zone, n = 7 mice per group. (**J**) Illustration of footshock-induced freezing behavior with optogenetic inhibition via a 589 nm laser. (**K**) Quantification of the freezing behavior, n = 7 mice per group. *p<0.05, all data are represented as mean ± SEM. Unpaired Student's t-test for (**F**), (**G**), (**I**), and (**K**).

The online version of this article includes the following figure supplement(s) for figure 4:

**Figure supplement 1.** Virus expression in the parabrachial nucleus (PBN) and the optic fiber position in the paraventricular nucleus (PVT) of Vglut2-ires-Cre mice injected with AAV2/9-EF1α-DIO-NpHR3.0-EYFP or AAV2/8-EF1α-DIO-EGFP.

**Figure supplement 2.** Optogenetic inhibition of parabrachial nucleus-paraventricular nucleus (PBN-PVT) projections did not affect associative fear memory acquisition and retrieval.

**Figure supplement 3.** Optogenetic inhibition of the parabrachial nucleus-paraventricular nucleus (PBN-PVT) neurons reduced aversion-like behavior and fear-like freezing behavior.

---

(*Figure 4H–I*). In addition to 2-MT, footshock is another paradigm that induces robust freezing behaviors. We found that constant inhibition of PBN-PVT projections reduced footshock-induced freezing behaviors (*Figure 4J and K*).

We also examined whether inhibition of the PBN-PVT projection affects aversive memory acquisition or retrieval (*Figure 4—figure supplement 2A*). We briefly suppressed the activity of PBN-PVT projections during footshock stimulation and found that freezing levels during the condition were not changed (*Figure 4—figure supplement 2B*). We further compared freezing levels in contextual and cue tests without or with laser and found that aversive memory retrieval was not affected either

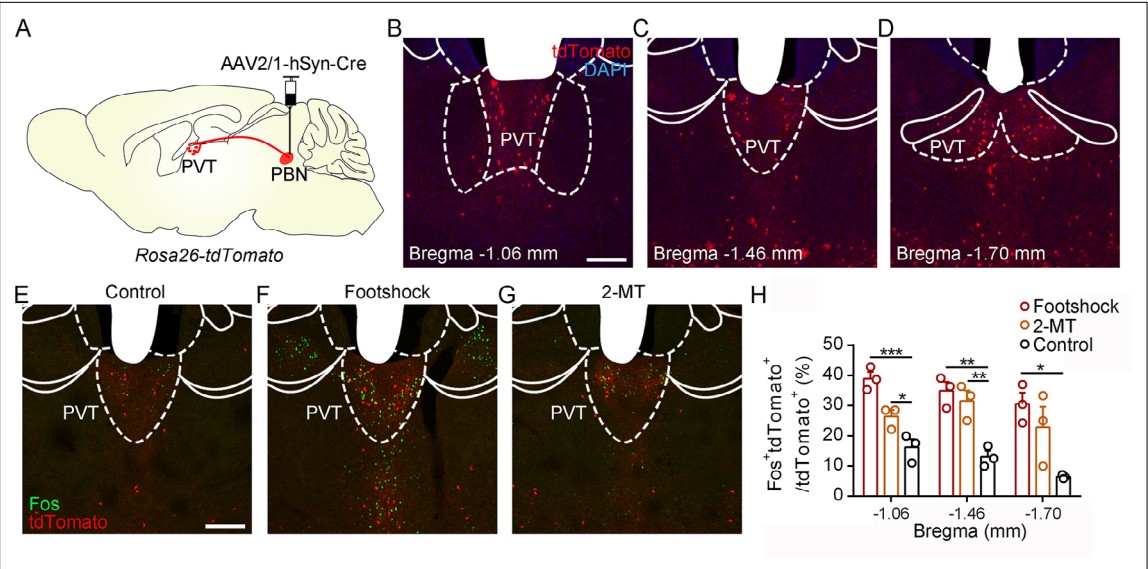

**Figure 5.** Activation of PVT_PBN neurons by diverse aversive stimuli. (**A**) The illustration shows the injection of AAV2/1-hSyn-Cre into the parabrachial nucleus (PBN) of *Rosa26-tdTomato* mice. (**B–D**) The distribution of the neurons in the paraventricular nucleus (PVT) at bregma –1.06 mm (**B**), bregma –1.46 mm (**C**), and bregma –1.70 mm (**D**). Scale bar: 200 µm. (**E–G**) Fos induced by habituation control (**E**), footshock (**F**), or 2-methyl-2-thiazoline (2-MT) (**G**) co-labeled with the tdTomato-positive neurons in the PVT. Scale bar: 200 µm. (**H**) Quantification of the co-labeled neurons, n = 3 mice per group. *p<0.05, **p<0.01, ***p<0.001, all data are represented as mean ± SEM, one-way ANOVA followed by Bonferroni test for (**H**).

The online version of this article includes the following figure supplement(s) for figure 5:

**Figure supplement 1.** Calcium signals of paraventricular nucleus (PVT) neurons in response to aversive stimuli.

(*Figure 4—figure supplement 2C and D*). In addition, we performed optogenetic inhibition of PBN-PVT neurons and observed similar phenomena (*Figure 4—figure supplement 3*). Thus, these results indicate that inhibition of the PBN-PVT pathway reduces aversion-like behaviors and footshock-induced freezing behaviors without affecting aversive memory acquisition or retrieval.

## PBN input shapes PVT neuronal responses to aversive stimulation

To further examine the activity of the PVT in response to aversive stimulation, we performed the in vivo fiber photometry and found that calcium signals of PVT neurons were increased after the foot-shock and air puff (*Figure 5—figure supplement 1*). In addition, we injected AAV2/1-hSyn-Cre virus, which could anterogradely label downstream neurons (*Zingg et al., 2017*), into the PBN of *Rosa26-tdTomato* mice (*Figure 5A*). The distribution pattern of PVT neurons that received PBN-PVT projection fibers (hereafter referred to as PVT_PBN neurons) is shown in *Figure 5B–D*. We used Fos as a marker to assess the activity change in 2-MT-treated mice and footshock-treated mice. The percentage of Fos+tdTomato+ neurons/tdTomato+ neurons in the PVT was significantly increased in the mice treated with aversive stimuli compared with that of control mice (*Figure 5E–H*), confirming that the PVT_PBN neurons could be activated by aversive stimuli.

The next question is whether PBN-PVT projections modulate the neuronal activity of PVT neurons in response to aversive stimuli. We first injected AAV2/9-EF1α-DIO-ChR2-mCherry virus into the PBN and performed dual Fos staining (*Nakahara et al., 2020*), detecting *fos* mRNA and Fos protein induced by two episodes of stimulation (*Figure 6—figure supplement 1A*). We found that there was a broad overlap between optogenetic stimulation-activated neurons (expressing the Fos protein) and footshock-activated neurons (expressing the *fos* mRNA) (*Figure 6—figure supplement 1B–E*).

Then, we injected AAV2/9-EF1α-DIO-ChR2-mCherry virus into the PBN and implanted the opto-electrode into the PVT of Vglut2-ires-Cre mice (*Figure 6A*). We first recorded the spiking signals in response to 10 sweeps of 2 s laser pulse trains (20 Hz, 5 mW, 5 ms). Then, we recorded the spiking signals in response to 20 sweeps of 2 s footshock (0.5 mA) without laser in the odd number sweeps or with laser in the even number sweeps (*Figure 6A*). We found that laser or footshock (without laser) increased firing rates in 22 or 28, respectively, of 40 neurons (*Figure 6B and C*). There was also a

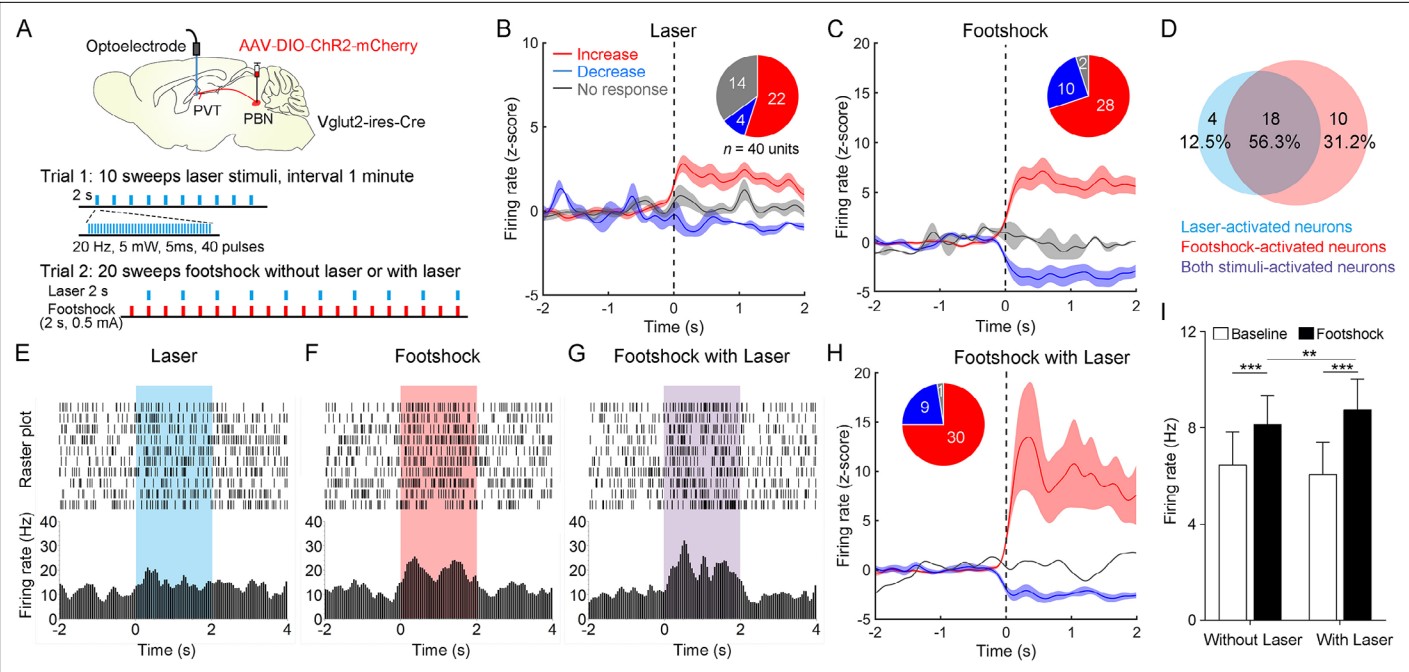

**Figure 6.** Neuronal activity of paraventricular nucleus (PVT) neurons in response to the footshock was modulated by parabrachial nucleus (PBN)-PVT projections. (**A**) Top: schematic shows injection of AAV2/9-EF1α-DIO-ChR2-mCherry into the PBN and placement of the optoelectrode above the PVT of Vglut2-ires-Cre mice. Bottom: the protocol of 10 sweeps of laser stimuli (Trial 1) and 20 sweeps of footshock stimuli without or with laser (Trial 2). (**B**) Firing rates (z-score) of 40 neurons during laser stimuli (20 Hz, 5 mW, 5 ms, 2 s). Inset: percentages of different groups of neurons according to z-score. (**C**) Firing rates (z-score) of 40 neurons during footshock (0.5 mA, 2 s) without laser stimuli. (**D**) Percentage of laser-activated, footshock-activated, and both stimuli-activated neurons. (**E–G**) Rastergrams and firing rates show the spiking activity of one PVT neuron during laser stimulus (**E**), footshock without laser stimulus (**F**), and footshock with laser stimulus (**G**). (**H**) Firing rates (z-score) of 40 neurons during footshock (0.5 mA, 2 s) with laser stimuli (20 Hz, 5 mW, 5 ms, 2 s). (**I**) Quantification of the firing rates of 40 neurons before and during footshock without and with laser, n = 40 neurons. **p<0.01, ***p<0.001, all data are represented as mean ± SEM, two-way ANOVA followed by Bonferroni test for (**I**).

The online version of this article includes the following figure supplement(s) for figure 6:

**Figure supplement 1.** Dual Fos staining detecting Fos protein and *fos* mRNA induced by laser stimulation and footshock.

**Figure supplement 2.** The response latency of paraventricular nucleus (PVT) neurons to laser activation and footshock activation.

broad overlap between laser-activated and footshock-activated neurons (*Figure 6D*). This result was consistent with the dual Fos staining result, suggesting that PVT_PBN neurons were activated by aversive stimulation. We also analyzed the response latency of laser-activated neurons and footshock-activated neurons (*Figure 6—figure supplement 2A–D*). The median response latency of 22 laser-activated neurons was 10 ms, suggesting that monosynaptic inputs from PBN could increase the PVT activity.

Next, we analyzed the firing rates of PVT neurons during footshock with laser sweeps and footshock without laser sweeps (*Figure 6E–G*). We found that the footshock stimulus laser-activated 30 of 40 neurons (*Figure 6H*) and increased the overall firing rates of 40 neurons compared with the footshock without laser result (*Figure 6I*). These results indicate that activation of PBN-PVT projections could enhance PVT neuronal responses to aversive stimulation.

## Pharmacogenetic activation of PVT_PBN neurons induces anxiety-like behaviors

We next investigate the functional role of PVT_PBN neurons in modulating negative affective states. We injected AAV2/1-hSyn-Cre virus bilaterally into the PBN and injected AAV2/9-hSyn-DIO-hM3Dq-mCherry virus or control virus into the PVT to activate PVT_PBN neurons (*Figure 7A*). The majority of the PVT_PBN neurons could be activated by CNO in the hM3Dq-expressing mice but not the control mice, as demonstrated by the Fos staining (*Figure 7B–D*). We found that pharmacogenetic activation of PVT_PBN neurons reduced the center time (*Figure 7E and F*). Similarly, the time spent in open quadrants was decreased and overall immobility time was increased in the EZM after activation of PVT_PBN neurons

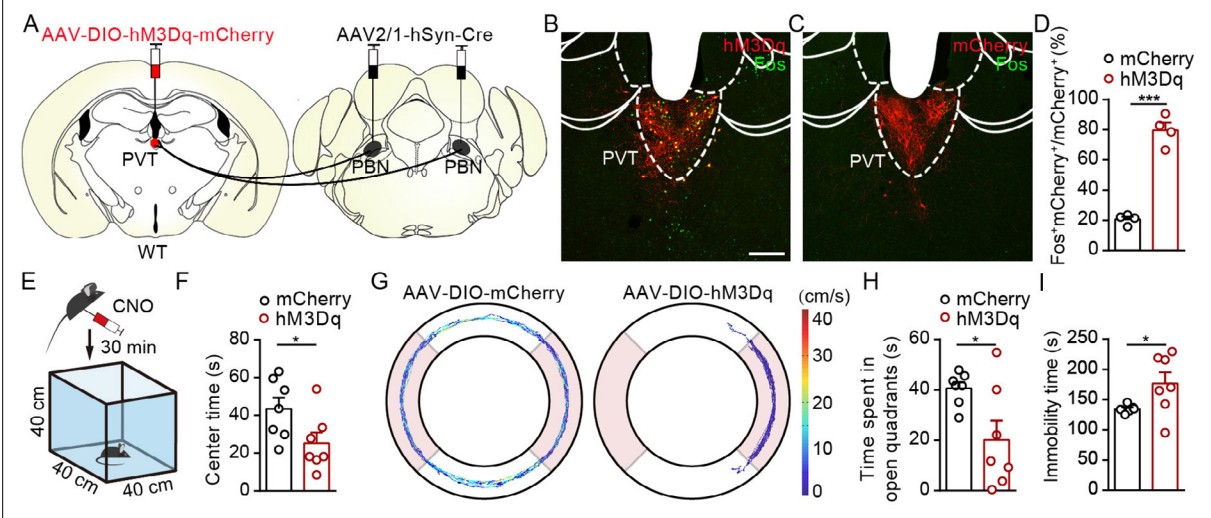

**Figure 7.** Activation of PVT$_{PBN}$ neurons induced anxiety-like behaviors. (**A**) The illustration shows injection of AAV2/1-hSyn-Cre into the parabrachial nucleus (PBN) and AAV2/9-hSyn-DIO-hM3Dq-mCherry into the paraventricular nucleus (PVT). (**B, C**) Clozapine N-oxide dihydrochloride (CNO) administration evoked Fos expression in AAV2/9-hSyn-DIO-hM3Dq-mCherry-injected mice (**B**) but not in AAV2/9-EF1α-DIO-mCherry-injected mice (**C**). Scale bar: 200 μm. (**D**) Percentage of co-labeled neurons in the PVT, n = 4 mice per group. (**E**) The illustration of the open field test (OFT) with pharmacogenetic activation. (**F**) Quantification of center time in the OFT, n = 7 mice per group. (**G**) Example of elevated zero maze (EZM) traces from the mice injected with AAV2/9-EF1α-DIO-mCherry or AAV2/9-hSyn-DIO-hM3Dq-mCherry. (**H, I**) Quantification of the time spent in open quadrants (**H**) and the immobility time in the EZM test (**I**), n = 7 mice per group. *p<0.05, ***p<0.001, all data are presented as mean ± SEM. Unpaired Student's *t*-test for (**D**), (**F**), (**H**), and (**I**).

The online version of this article includes the following figure supplement(s) for figure 7:

**Figure supplement 1.** Distribution pattern of projection fibers of PVT$_{PBN}$ neurons.

---

(*Figure 7G–I*). We did not observe obvious nociception-related behaviors, such as forelimb wiping, hindlimb flinching, licking, or biting, after activation of PVT$_{PBN}$ neurons. These results indicate that pharmacogenetic activation of PVT$_{PBN}$ neurons induces anxiety-like behaviors.

Furthermore, we examined the anatomic distribution of terminals of PVT$_{PBN}$ neurons. We labeled PVT$_{PBN}$ neurons in WT mice by injecting AAV2/1-hSyn-Cre virus into the PBN and AAV2/8-EF1α-DIO-EGFP virus into the PVT (*Figure 7—figure supplement 1A and B*). We found that PVT$_{PBN}$ neurons sent projections to several brain areas, in particular the NAc, BNST, and CeA (*Figure 7—figure supplement 1B–H*), which was similar to early tracing research on PVT efferent projections (*Li and Kirouac, 2008*).

## Discussion

In this study, we employed viral tracing and electrophysiology to confirm the monosynaptic excitatory connectivity between the PBN and the PVT. Optogenetic or pharmacogenetic activation of the PBN-PVT pathway induced anxiety-like, aversion-like, and fear-like behaviors. Optogenetic inhibition of the PBN-PVT pathway could partially reduce 2-MT-induced aversive behaviors as well as footshock-induced freezing behaviors. The activity of PVT neurons was increased with the presentation of several aversive stimuli and could be further increased by activation of PBN-PVT projections. Consistently, activation of PVT$_{PBN}$ neurons induced anxiety-like behaviors. Taken together, our results reveal the functional role of PBN-PVT projections in modulating negative affective states in mice.

### PBN efferents and PBN-PVT monosynaptic excitatory projections

The PBN is a critical hub receiving sensory information from the spinal cord (*Todd, 2010*). The widespread distribution of PBN efferents contributes to different aspects of behavioral and physiological responses. Previous studies have shown that CGRP-expressing neurons in the PBN project to the CeA and contribute to the affective dimension of pain. In contrast, non-CGRP neurons may transmit sensory pain information (*Han et al., 2015*). The projections from the PBN to the VMH or PAG are

involved in producing escape behaviors to avoid injury, while the projections from the PBN to the BNST or CeA participate in facilitating aversive memory (*Chiang et al., 2020*). PBN neurons, which receive projections from the spinal cord, form strong functional synaptic connections with ILN neurons but not CeA neurons to process nociceptive signals (*Deng et al., 2020*). The PVT, which is located in the midline of the brain, is an important area that participates in the processing of affective states (*Kirouac, 2015*). Although recent research has reported that projecting fibers from the PBN were found in the PVT (*Chiang et al., 2020*), remarkably little is known about the connectivity information and function of these PBN-PVT projections.

Since we injected the constitutively expressed ChR2 virus into the PBN, some neurons in the LC (which is medial to the PBN) might be infected. LC neurons express type 1 vesicular glutamate transporter (VgluT1) and also project to the PVT (*Beas et al., 2018*). Although PBN-PVT projections comprise the major portion of the projections, there is still potential contamination from LC-PVT projections. We also observed a small portion of inhibitory connections between the PBN and the PVT by performing slice recording experiments. These results are consistent with a previous study showing that GABAergic neurons in the PBN also send sparse projections to the PVT (*Chiang et al., 2020*). We speculate that this PBN-PVT GABAergic pathway might play an opposite role in modulating negative affective states. Further in situ hybridization results confirmed that the PBN-PVT neurons were mainly glutamatergic neurons expressing *Slc17a6* mRNA. These results suggest that the majority of the PBN-PVT projections appeared to be excitatory. Further optoelectrode experiment results also confirmed that firing rates of some PVT neurons could be directly increased by monosynaptic inputs from PBN.

We also found that the connectivity and density of glutamatergic PBN projection fibers were higher in the middle and pPVT. These results are consistent with various studies supporting the idea that the pPVT is a particularly sensitive region of the PVT to aversive stimuli (*Gao et al., 2020*; *Beas et al., 2018*; *Barson et al., 2020*).

## PBN-PVT projections modulate negative affective states

We found that activation of PBN-PVT projections or PBN-PVT neurons induced anxiety-like behaviors and fear-like behaviors in the OFT and EZM. We observed that mice displayed robust running and jumping behaviors mainly in the first minute of optogenetic manipulation, and these phenomena were not observed in the pharmacogenetic experiment. These might have been caused by the instantly increased activity of PBN-PVT projections induced by optogenetic manipulation. Mice might display 'fight or flight' during sudden affective state transitions. The pharmacogenetic approach takes several minutes and gradually enhances neural activity, and the resulting affective states change in a relatively mild way. We also observed that anxiety-like behaviors in the OFT still existed several minutes after optogenetic activation of PBN-PVT projections. However, in the RTPA test, aversion appeared when the laser was on and disappeared when the laser was off, indicating that aversion was transient and could not be translated to associative learning. This was further confirmed by the prolonged CPA test and cue-dependent optogenetic conditioning test. These results suggest that activation of PBN-PVT projections induces instant negative affective states but does not drive associative fear memory formation.

The selective optogenetic inhibition of PBN-PVT projections or PBN-PVT neurons could reduce aversion-like and fear-like behaviors. To better examine the behavioral changes, we performed a 10 min test in 2-MT and footshock experiments. We used a relatively long-term protocol in optogenetic inhibition experiments (10 min of constant laser stimulation). Such long-term inhibition protocols have been used in other studies (*Zhou et al., 2019*; *Sun et al., 2020*). We also performed the classic fear conditioning test and found that inhibition of PBN-PVT projections did not affect associative fear memory formation or retrieval, suggesting that PBN-PVT projections mainly promote aversion but do not facilitate negative associations.

Our calcium imaging and Fos staining results indicate that PVT neurons are activated after exposure to aversive stimuli, consistent with a previous study (*Zhu et al., 2018*). The dual Fos staining experiment and optoelectrode experiments confirmed a broad overlap between laser-activated and footshock-activated neurons. The median response latency of laser-activated PVT neurons was 10 ms, indicating that monosynaptic inputs from PBN could directly activate a part of PVT neurons. We also observed that some neurons had relatively long latency (from 20 to 150 ms). That might be caused by cumulative effects of monosynaptic inputs or antidromic activation of PBN and multisynaptic

activation of other PVT inputs that are innervated by the PBN. Further analysis showed that activation of PBN-PVT projections enhanced the overall firing rates of PVT neurons in response to footshock. These results suggest that the activation of PBN-PVT projections could enhance neuronal activity in response to aversive stimulation.

Previous studies have reported that activation of the PBN-CeA pathway was sufficient to drive behaviors associated with negative affective states (*Bowen et al., 2020*; *Cai et al., 2018*; *Han et al., 2015*), enable associative learning, and generate aversive memory (*Chiang et al., 2020*). Distinct from PBN-CeA projections, we found that activation of PBN-PVT projections induced only transient aversion-related behaviors, and inhibition of PBN-PVT projections did not affect fear memory acquisition or retrieval. A study reported that only a few fluoro-gold (FG)/tetramethylrhodamine-dextran (TMR) double-labeled neurons were sparsely distributed in the PBN of mice injected with FG into the PVT and TMR into the CeA (*Liang et al., 2016*). Our results also showed few collateral projecting fibers in the CeA or VMH from PBN-PVT neurons. These results suggested that the PBN-PVT pathway and the PBN-CeA pathway are two parallel pathways originating from distinct efferent neurons within the PBN that perform distinct functions.

Recently, Chiang et al. found two major efferent pathways from the lateral PBN: one originating from the dorsal division of lateral PBN collateralizes the VMH and PAG and a second arising from the external lateral division of lateral PBN that collateralize to the BNST and CeA. They suggested that activation of the first pathway generates the aversive memory, and activation of the second one drives escape behaviors (*Chiang et al., 2020*). In our study, the results indicate that the PBN-PVT pathway arises from both dorsal and external lateral divisions of lateral PBN and collateralizes the BNST, LH, PVN, and PAG but not the CeA or VMH. According to the location of originated PBN neurons and collateral projection pattern, we speculated that the PBN-PVT efferent pathway is different from both PBN-VMH/PAG pathway and PBN-BNST/CeA pathway. Since there were broad collateral projection fibers in BNST, PAG, and other brain regions, the possibility of antidromic effects following photoactivation of PBN terminals in PVT should be considered.

The tracing results showed that PVT$_{PBN}$ neurons projected to multiple brain areas, particularly the NAc, BNST, and CeA. The BNST and CeA have been previously implicated in negative affective behaviors (*Jennings et al., 2013*; *Tye et al., 2011*). Previous studies have shown that the activation of PVT-CeA projections induced place aversion, and the effect persisted the next day in the absence of photostimulation (*Do-Monte et al., 2017*). Similarly, long-term depression (LTD)-like stimulation of PVT-CeA projections or inhibition of the same circuit induced a persistent attenuation of fear responses (*Chen and Bi, 2019*; *Do-Monte et al., 2015*; *Penzo et al., 2015*). These results revealed a critical role of PVT-CeA projections in aversive memory formation. In our study, we found that PBN-PVT projections were not crucial for aversive memory formation. The possible reason might be that manipulation of PVT-CeA projections induced direct excitatory inputs to the CeA, and the inputs were strong enough for aversive memory formation. However, activation of PBN-PVT projections might not induce enough excitatory inputs to the CeA via the disynaptic connection.

A previous study also found that the PVT mediates descending pain facilitation underlying persistent pain conditions via the PVT-CeA-PAG circuit (*Liang et al., 2020*). Different downstream pathways of PVT$_{PBN}$ neurons might have different functions, and deciphering the circuit mechanisms needs further examination.

## The potential role of PBN-PVT projections in depression and pain

It is worth noting that although the pharmacogenetic activation of PBN-PVT neurons induced anxiety-like behaviors and fear-like behaviors in hM3Dq group mice, no depression-like symptoms were observed in the TST and FST. On the other hand, chronic pain models, such as the partial sciatic nerve ligation model, spared nerve injury model, and complete Freund's adjuvant model, generally induce anxiety and depression at least 3–4 weeks after surgery in mice (*Dimitrov et al., 2014*; *Zhou et al., 2019*). Our study collected behavioral data 30 min after a single dose of CNO injection. Different behavioral tests were performed at least 3 days apart to eliminate the residual CNO effects. We hypothesized that depression-like behaviors might be observed if we repeatedly activated PBN-PVT projections for weeks. However, whether PBN-PVT projections are involved in depression is still unknown.

A recent study revealed that PBN neurons convey nociceptive information from the spinal cord to the ILN, which is relatively close to the PVT (*Deng et al., 2020*). In our results, we carefully checked the virus expression and optic fiber locations. We found that pharmacogenetic activation of PBN-PVT neurons did not affect basal nociceptive thresholds or formalin-induced licking behaviors. Moreover, no obvious nociception-related behaviors were found through specific manipulations of the PBN innervated PVT neurons, which suggests that PBN-PVT projections might not be involved in nociceptive information processing.

In summary, we identified the functional role of PBN-PVT projections in modulating negative affective states. Our study paves the way for further deciphering the distinct roles of the PBN neural circuit in affective behaviors.

# Materials and methods

## Key resources table

| Reagent type (species) or resource | Designation | Source or reference | Identifiers | Additional information |
|---|---|---|---|---|
| Genetic reagent (*Mus musculus*) | *B6.Cg-Gt(ROSA)26Sortm9(CAG-tdTomato)Hze/J (Ai9)* | Jackson Laboratory | Stock# 007909; RRID:MGI:3813511 | Dr. Hua-Tai Xu (Institutes of Neuroscience, Chinese Academic of Sciences) |
| Genetic reagent (*M. musculus*) | *STOCK Slc17a6tm2(cre)Lowl/J* (Vglut2-ires-Cre) | Jackson Laboratory | Stock# 016963; RRID:MGI:5300532 | Dr. Yan-Gang Sun (Institutes of Neuroscience, Chinese Academic of Sciences) |
| Genetic reagent (*Dependoparvovirus*) | AAV2/8-hSyn-ChR2-mCherry | Obio Technology | Cat# AG26976 | $4 \times 10^{12}$ v.g./mL |
| Genetic reagent (*Dependoparvovirus*) | AAV2/8-EF1α-DIO-EGFP | Taitool Bioscience | Cat# S0270 | $4 \times 10^{12}$ v.g./mL |
| Genetic reagent (*Dependoparvovirus*) | retroAAV2/2-hSyn-Cre | Taitool Bioscience | Cat# S0278-2RP-H20 | $4 \times 10^{12}$ v.g./mL |
| Genetic reagent (*Dependoparvovirus*) | AAV2/9-EF1α-DIO-ChR2-mCherry | Taitool Bioscience | Cat# S0170-9-L20 | $4 \times 10^{12}$ v.g./mL |
| Genetic reagent (*Dependoparvovirus*) | AAV2/9-EF1α-DIO-mCherry | Obio Technology | Cat# AG20299 | $4 \times 10^{12}$ v.g./mL |
| Genetic reagent (*Dependoparvovirus*) | AAV2/9-hSyn-DIO-hM3Dq-mCherry | BrainVTA | Cat# PT-0019 | $4 \times 10^{12}$ v.g./mL |
| Genetic reagent (*Dependoparvovirus*) | AAV2/9-EF1α-DIO-NpHR3.0-EYFP | Obio Technology | Cat# AG26966 | $4 \times 10^{12}$ v.g./mL |
| Genetic reagent (*Dependoparvovirus*) | AAV2/8-hSyn-GCaMP6s | Taitool Bioscience | Cat# S0225-8 | $4 \times 10^{12}$ v.g./mL |
| Genetic reagent (*Dependoparvovirus*) | AAV2/1-hSyn-Cre | Taitool Bioscience | Cat# S0278-1-H50 | $1.5 \times 10^{13}$ v.g./mL |
| Sequence-based reagent | RNAscope Probe-*fos*-C2 | Advanced Cell Diagnostics | Cat# 316921-C2 | |
| Sequence-based reagent | RNAscope Probe-*Tac1*-C2 | Advanced Cell Diagnostics | Cat# 410351-C2 | |
| Sequence-based reagent | RNAscope Probe-*Tacr1*-C2 | Advanced Cell Diagnostics | Cat# 428781-C2 | |
| Sequence-based reagent | RNAscope Probe-*Pdyn* | Advanced Cell Diagnostics | Cat# 318771 | |
| Sequence-based reagent | RNAscope Probe- *Slc17a6*-C2 | Advanced Cell Diagnostics | Cat# 319171-C2 | |
| Antibody | Anti-Fos (rabbit polyclonal) | Abcam | Cat# ab190289; RRID:AB_2737414 | IF (1:4000) |
| Antibody | Anti-CGRP (goat polyclonal) | Abcam | Cat# ab36001; RRID:AB_725807 | IF (1:1000) |
| Antibody | Anti-DsRed (goat polyclonal) | Takara Bio | Cat# 632496; RRID:AB_10013483 | IF (1:500) |
| Antibody | Alexa Fluor 488 AffiniPure Donkey Anti-Rabbit IgG (H+L) | Jackson ImmunoResearch Labs | Cat# 711-545-152; RRID:AB_2313584 | IF (1:400) |

*Continued on next page*

*Continued*

| Reagent type (species) or resource | Designation | Source or reference | Identifiers | Additional information |
|---|---|---|---|---|
| Antibody | Cy3 AffiniPure Donkey Anti-Rabbit IgG (H+L) | Jackson ImmunoResearch Labs | Cat# 711-165-152; RRID:AB_2307443 | IF (1:400) |
| Antibody | Alexa Fluor 488 AffiniPure F(ab')₂ Fragment Donkey Anti-Goat IgG (H+L) | Jackson ImmunoResearch Labs | Cat# 705-546-147; RRID:AB_2340430 | IF (1:400) |
| Commercial assay or kit | RNAscope Multiplex Fluorescent Reagent Kit v2 | Advanced Cell Diagnostics | Cat# 320293 | |
| Software, algorithm | ImageJ | NIH | | |
| Software, algorithm | LabState | AniLab | | |

## Animals

Male C57Bl/6J wild-type mice, *Rosa26-tdTomato* mice (Jax Stock# 007909, gifted from Dr. Hua-Tai Xu, Institutes of Neuroscience, Chinese Academic of Sciences), and Vglut2-ires-Cre mice (Jax Stock# 016963, gifted from Dr. Yan-Gang Sun, Institutes of Neuroscience, Chinese Academic of Sciences) were used. Animals were housed in standard laboratory cages in a temperature (23–25°C)-controlled vivarium with a 12:12 light/dark cycle, free to food and water. For tracing and behavioral experiments, the mice were injected with the virus at 7–8 weeks old and performed the behavioral tests at 11–12 weeks old. For the electrophysiological experiments, the mice were injected with the virus at 4–6 weeks old to accomplish the electrophysiological experiments at 7–9 weeks old. For in vivo fiber photometry and optoelectrode experiments, the mice were injected with the virus at 7–8 weeks old to accomplish the experiments at 10–11 weeks old. All animal experiment procedures were approved by the Animal Care and Use Committee of Shanghai General Hospital (2019AW008).

## Stereotaxic surgery

Mice were anesthetized by vaporized sevoflurane (induction, 3%; maintenance, 1.5%) and head-fixed in a mouse stereotaxic apparatus (RWD Life Science Co.).

For electrophysiological experiments, the AAV2/8-hSyn-ChR2-mCherry virus (300 nL, 4 × 10¹² v.g./mL, AG26976, Obio Technology) was injected into the PBN nucleus of WT mice in the stereotaxic coordinate: anteroposterior (AP) −5.2 mm, mediolateral (ML) +1.3 mm, and dorsoventral (DV) −3.4 mm.

For tracing studies, the AAV2/8-EF1α-DIO-EGFP virus (300 nL, S0270, Taitool Bioscience) was injected into the PBN (mentioned above) of Vglut2-ires-Cre mice.

For the retrovirus injection surgery, the retrograde transport Cre recombinase retroAAV2/2-hSyn-Cre virus (150 nL, 4 × 10¹² v.g./mL, S0278-2RP-H20, Taitool Bioscience) was injected in the *Rosa26-tdTomato* mice at two locations of PVT, respectively: (1) AP −1.22 mm, ML 0 mm, DV −2.9 mm; (2) AP −1.46 mm, ML 0 mm, DV −2.9 mm.

For optogenetic activation of PVT-projecting PBN fibers, the AAV2/9-EF1α-DIO-ChR2-mCherry virus (300 nL, 4 × 10¹² v.g./mL, S0170-9-L20, Taitool Bioscience) or the AAV2/9-EF1α-DIO-mCherry virus (300 nL, 4 × 10¹² v.g./mL, AG20299, Obio Technology) was bilaterally injected into the PBN (mentioned above) of Vglut2-ires-Cre mice, and a 200 μm diameter optic fiber was implanted over the PVT (AP −1.46 mm, ML 0 mm, DV −2.9 mm) with a 20° angle towards the midline.

For the pharmacogenetic activation of PBN-PVT neurons, the retroAAV2/2-hSyn-Cre virus (150 nL, 4 × 10¹² v.g./mL, S0278-2RP-H20, Taitool Bioscience) was injected into the PVT (AP −1.46 mm, ML 0 mm, DV −2.9 mm), and the AAV2/9-hSyn-DIO-hM3Dq-mCherry virus (300 nL, 4 × 10¹² v.g./mL, PT-0019, BrainVTA) or the control AAV2/9-EF1α-DIO-mCherry virus was bilaterally injected into the PBN (mentioned above) of the WT mice.

For optogenetic inhibition of PVT-projecting PBN fibers, AAV2/9-EF1α-DIO-NpHR3.0-EYFP virus (300 nL, 4 × 10¹² v.g./mL, AG26966, Obio Technology) or the AAV2/8-EF1α-DIO-EGFP virus was bilaterally injected into the PBN (mentioned above) of Vglut2-ires-Cre mice, and a 200 μm diameter optic fiber was implanted over the PVT (AP −1.46 mm, ML 0 mm, DV −2.9 mm) with a 20° angle towards the midline.

For optogenetic inhibition of PBN-PVT neurons, retroAAV2/2-hSyn-Cre was injected into the PVT (AP −1.46 mm, ML 0 mm, DV −2.9 mm), and AAV2/9-EF1α-DIO-NpHR3.0-EYFP virus (300 nL,

4 × 10$^{12}$ v.g./mL, AG26966, Obio Technology) or the AAV2/8-EF1α-DIO-EGFP virus was bilaterally injected into the PBN of WT mice, the left optic fiber was implanted over the PBN vertically, and the right one was placed over the PBN with a 20° angle towards the midline.

For in vivo fiber photometry experiments, the AAV2/8-hSyn-GCaMP6s virus (200 nL, 4 × 10$^{12}$ v.g./mL, S0225-8, Taitool Bioscience) was injected into the PVT nucleus (AP −1.46 mm, ML 0 mm, DV −2.90 mm) of the WT mice, and the optic fiber was implanted above the PVT with a 20° angle towards the midline.

For optoelectrode experiments, the AAV2/9-EF1α-DIO-ChR2-mCherry virus (300 nL, 4 × 10$^{12}$ v.g./mL, S0170-9-L20, Taitool Bioscience) was bilaterally injected into the PBN (mentioned above) of Vglut2-ires-Cre mice. Three weeks later, the homemade optoelectrode was implanted into the PVT nucleus (AP −1.46 mm, ML 0 mm, DV −2.90 mm).

For pharmacogenetic activation of PVT$_{PBN}$ neurons, the AAV2/1-hSyn-Cre virus (300 nL, 1.5 × 10$^{13}$ v.g./mL, S0278-1-H50, Taitool Bioscience) was bilaterally injected into the PBN nucleus, and the AAV2/9-hSyn-DIO-hM3Dq-mCherry virus or the control AAV2/9-EF1α-DIO-mCherry virus was injected into the PVT (AP −1.46 mm, ML 0 mm, DV −2.9 mm) of the WT mice.

The virus was infused through a glass pipette (10–20 μm in diameter at the tip) at the rate of 50–100 nL/min. The injection pipette was left in place for additional 8 min. After the surgeries, the skin was closed by the sutures, and the optic fiber was secured through the dental acrylic. Generally, tracing, electrophysiological, or behavioral experiments were performed at least 3 weeks later. After experiments, histological analysis was used to verify the location of viral transduction and the optic fiber. The mice without correct transduction of virus or correct site of optic fiber were excluded for analysis.

## Histology

Animals were deeply anesthetized with vaporized sevoflurane and transcardially perfused with 20 mL saline, followed by 20 mL paraformaldehyde (PFA, 4% in PBS). Brains were extracted and soaked in 4% PFA at 4°C for a minimum of 4 hr and subsequently cryoprotected by transferring to a 30% sucrose solution (4°C, dissolved in PBS) until brains were saturated (for 36–48 hr). Coronal brain sections (40 μm) were cut using a freezing microtome (CM1950, Leica). The slices were collected and stored in PBS at 4°C until immunohistochemical processing. Nuclei were stained with DAPI (Beyotime, 1:10000) and washed three times with PBS.

The brain sections undergoing immunohistochemical staining were washed in PBS three times (10 min each time) and incubated in a blocking solution containing 0.3% TritonX-100 and 5% normal donkey serum (Jackson ImmunoResearch, USA) in PBS for 1 hr at 37°C. Sections were then incubated (4°C, 24 hr) with primary antibodies dissolved in 1% normal donkey serum solution. Afterward, sections were washed in PBS four times (15 min each time), then incubated with secondary antibodies for 2 hr at room temperature. After DAPI staining and washing with PBS, sections were mounted on glass microscope slides, dried, and covered with 50% glycerin (Thermo Fisher). The images were taken with the Leica Dmi8 microscope and by the Leica SP8 confocal microscopy. The images were further processed using Fiji and Photoshop.

## RNAscope in situ hybridization

Mice were anesthetized with isoflurane and rapidly decapitated. Brains were roughly dissected from perfused mice and post-fixed in 4% PFA at 4°C overnight, dehydrated in 30% sucrose 1× PBS at 4°C for 2 days. Mouse brains were embedded, cryosectioned in 15 μm coronal slices, and mounted on SuperFrost Plus Gold slides (Fisher Scientific). In situ hybridization was performed according to the protocol of the RNAscope Multiplex Fluorescent Reagent Kit v2 (Cat# 320293). Probes were purchased from Advanced Cell Diagnostics: *fos* (Cat# 316921-C2), *Tac1* (Cat# 410351-C2), *Tacr1* (Cat# 428781-C2), *Pdyn* (Cat# 318771), and *Slc17a6* (Cat# 319171-C2). Primary antibodies include rabbit anti-Fos (Abcam, Cat# ab190289, 1:4000), goat anti-CGRP (Abcam, Cat# ab36001, 1:1000), and rabbit anti-DsRed (Takara Bio, Cat# 632496, 1:500). All secondary antibodies were purchased from Jackson ImmunoResearch and used at 1:400 dilution. Secondary antibodies include Alexa 488 donkey anti-rabbit (Cat# 711-545-152), Cy3 donkey anti-rabbit (Cat# 711-165-152), and Alexa 488 donkey anti-goat (Cat# 705-546-147). Images were collected on a Leica fluorescence microscope and Leica LAS Software.

## Fos induction

The mice were habituated for 3 days and performed gentle grabbing and holding for 1 min, five times every day, to minimize background Fos expression.

To study the effect of pharmacogenetic manipulations on PBN-PVT neurons, we intraperitoneally injected 0.5 mg/kg CNO (Sigma). 90 min later, the brain tissues were processed.

To assess 2-MT-evoked Fos expression in the PVT, the mice were kept in a chamber with a floor covered with cotton containing 100 mL 1:1000 diluted 2-MT for 90 min. Then the mice were perfused.

To assess footshock-induced Fos expression in the PVT, we placed the mice into the chamber and delivered 30 times inevitable footshock (0.5 mA, 1 s) with a variable interval (averaging 60 s). After stimulation, animals were kept in the same apparatus for another 60 min, and brain tissues were then processed.

For the dual Fos experiments, we first delivered 20 min 473 nm laser pulses (20 Hz, 5 mW, 5 ms) and left the mice to rest in the homecage for 60 min. Then we delivered the 20 min footshock stimulus (0.5 mA, 1 s, 30 times, variable interval) and perfused the mice.

## Electrophysiology

The electrophysiological experiment was performed as previously described (*Mu et al., 2017*). Mice were anesthetized with sevoflurane and perfused by the ice-cold solution containing (in mM) 213 sucrose, 2.5 KCl, 1.25 $NaH_2PO_4$, 10 $MgSO_4$, 0.5 $CaCl_2$, 26 $NaHCO_3$, 11 glucose (300–305 mOsm). Brains were quickly dissected, and the coronal slices (250 µm) containing the PBN or PVT were chilled in ice-cold dissection buffer using a vibratome (V1200S, Leica) at a speed of 0.12 mm/s. The coronal sections were subsequently transferred to a chamber and incubated in the artificial cerebrospinal fluid (ACSF, 34°C) containing (in mM) 126 NaCl, 2.5 KCl, 1.25 $NaH_2PO_4$, 2 $MgCl_2$, 2 $CaCl_2$, 26 $NaHCO_3$, 10 glucose (300–305 mOsm) to recover for at least 40 min, then kept at room temperature before recording. All solutions were continuously bubbled with 95% $O_2$/5% $CO_2$.

All experiments were performed at near-physiological temperatures (30–32°C) using an in-line heater (Warner Instruments) while perfusing the recording chamber with ACSF at 3 mL/min using a pump (HL-1, Shanghai Huxi). Whole-cell patch-clamp recordings were made from the target neurons under IR-DIC visualization and a CCD camera (Retiga ELECTRO, QIMAGING) using a fluorescent Olympus BX51WI microscope. Recording pipettes (2–5 MΩ; Borosilicate Glass BF 150-86-10; Sutter Instrument) were prepared with a micropipette puller (P97; Sutter Instrument) and backfilled with potassium-based internal solution containing (in mM) 130 K-gluconate, 1 $MgCl_2$, 1 $CaCl_2$, 1 KCl, 10 HEPES, 11 EGTA, 2 Mg-ATP, 0.3 Na-GTP (pH 7.3, 290 mOsm), or cesium-based internal solution containing (in mM) 130 $CsMeSO_3$, 1 $MgCl_2$, 1 $CaCl_2$, 10 HEPES, 2 QX-314, 11 EGTA, 2 Mg-ATP, 0.3 Na-GTP (pH 7.3, 295 mOsm). Biocytin (0.2%) was included in the internal solution.

In PBN-PVT ChR2 experiments, whole-cell recordings of PBN neurons with current-clamp (I = 0 pA) were obtained with pipettes filled with the potassium-based internal solution. The 473 nm laser (5 Hz, 10 Hz, 20 Hz pulses, 0.5 ms duration, 2 mW/mm²) was used to activate PBN ChR2-positive neurons. Light-evoked EPSCs and IPSCs of PVT neurons recorded with voltage-clamp (holding voltage of –70 mV or 0 mV) were obtained with pipettes filled with the cesium-based internal solution. The 473 nm laser (20 Hz paired pulses, 1 ms duration, 4 mW/mm²) was used to activate ChR2-positive fibers. The light-evoked EPSCs were completely blocked by 1 µM TTX, rescued by 100 µM 4-AP, and blocked by 10 µM (6-nitro-7-sulphamoylbenzo(f)quinoxaline-2,3-dione (NBQX)). NBQX and TTX were purchased from Tocris Bioscience. All other chemicals were obtained from Sigma.

Voltage-clamp and current-clamp recordings were carried out using a computer-controlled amplifier (MultiClamp 700B; Molecular Devices, USA). During recordings, traces were low-pass filtered at 4 kHz and digitized at 10 kHz (DigiData 1550B1; Molecular Devices). Data were acquired by Clampex 10.6 and filtered using a low-pass Gaussian algorithm (–3 dB cutoff frequency = 1000 Hz) in Clampfit 10.6 (Molecular Devices).

## Optogenetic manipulation

For activating the PBN-PVT projection, a 473 nm laser (20 Hz, 5 mW, 5 ms pulse duration) was delivered. For inhibition of the PBN-PVT projection and the PBN-PVT neurons, a constant laser (589 nm, 10 mW) was delivered.

## Pharmacogenetic manipulation

All behavioral tests were performed 30 min after i.p. injection of 0.5 mg/kg CNO in pharmacogenetic manipulation. Different behavior tests were performed at least 3 days apart.

## Open field test

The OFT was used to assess locomotor activity and anxiety-related behavior in an open field arena (40 × 40 × 60 cm) with opaque plexiglass walls. The mouse was placed in the center of the box and recorded with a camera attached to a computer. The movement was automatically tracked and analyzed using AniLab software (Ningbo AnLai, China). The total distance traveled, the total velocity, the total immobility time (the mice were considered to be immobile if immobility time lasts more than 1 s), and time spent in the center area (20 × 20 cm) were measured. The box was cleaned with 70% ethanol after each trial.

To assess the effect of optogenetic activation of the PBN-PVT projection, 15 min sessions consisting of 5 min pre-test (laser OFF), 5 min laser on test (laser on), and 5 min post-test (laser off) periods. Laser (473 nm, 20 Hz, 5 mW, 5 ms) was delivered during the laser on phase.

To assess the effect of pharmacogenetic manipulations of PBN-PVT neurons on locomotor activity and affective behaviors, we recorded the movement 30 min after i.p. injection with CNO.

To assess the effect of inhibition of the PBN-PVT projection on the aversive behaviors induced by 2-MT, one cotton ball containing 5 mL 2-MT (1:1000) solution was placed on the center of the upper-left quadrant to disseminate fear-odor, then a constant laser (589 nm, 10 mW) was delivered during the 10 min test. The time spent in the 2-MT paired quadrant was calculated.

## Elevated zero maze

The EZM was an opaque plastic circle (60 cm diameter), which consisted of four sections with two opened and two closed quadrants. Each quadrant had a path width of 6 cm. The maze was elevated 50 cm above the floor. The animals were placed into an open section facing a closed quadrant and freely explored the maze for 5 min.

## Real-time place aversion test

Mice were habituated to a custom-made 20 × 30 × 40 cm two-chamber apparatus (distinct wall colors and stripe patterns) before the test. Each mouse was placed in the center and allowed to explore both chambers without laser stimulation for 10 min on day 1. The movement was recorded for 10 min as a baseline. The mice performed a slight preference for the black chamber according to the fact that the mice have innate aversion to brightly illuminated areas. On day 2, 473 nm laser stimulation (20 Hz, 5 mW, 5 ms) was automatically delivered when the mouse entered or stayed in the black chamber and turned off when the mouse exited the black chamber for 10 min. Finally, the mouse was allowed to freely explore both chambers without laser stimulation for another 10 min. The RTPA location plots and total time on the stimulated side were recorded and counted with the AniLab software.

## Conditioned place aversion

After habituation, mice were placed in the center of the two-chamber apparatus and allowed to explore either chamber for 15 min on day 1. On day 2, mice were restricted to one chamber (laser-paired chamber) with photostimulation (473 nm, 20 Hz, 5 mW, 5 ms) for 30 min in the morning and restricted to the other chamber (unpaired chamber) without photostimulation in the afternoon. On day 3, mice were restricted to the unpaired chamber without photostimulation in the morning and restricted to the laser-paired chamber with photostimulation in the afternoon. On day 4, mice were allowed to explore both chambers without laser stimulation for another 15 min. The time in the laser-paired chamber was calculated on days 1 and 4.

## 2-MT-induced aversion

To assess the effect of optogenetic inhibition of the PBN-PVT projection or PBN-PVT neurons on the aversive state, three cotton balls containing 15 mL 2-MT (1:1000) solution were placed in the black chamber. A constant laser (589 nm, 10 mW) was delivered during the 10 min test.

## Cue-dependent optogenetic conditioning test

Video Freeze fear conditioning system with optogenetic equipment (MED Associates, MED-VFC-OPTO-USB-M) and Video Freeze software were used.

On day 1, mice were habituated to the fear conditioning chambers and allowed to explore for 2 min freely, then three tones (75 dB, 4 kHz, 30 s duration) separated by a variable interval with a range of 60–120 s and the average of 90 s were delivered.

On day 2, mice were trained with the sound cue (75 dB, 4 kHz, 30 s) paired with a simultaneous 30 s laser pulse train (20 Hz, 5 ms, 5 mW) for six times separated by a variable interval (averaging 90 s). The mice were kept in the conditioning chamber for another 60 s before returning to the home cages.

On day 3, mice were placed back into the original training chamber for 3 min to perform the contextual test. After 2–3 hr, the conditioning chamber was modified by changing its metal floor and sidewalls. Mice were placed in the altered chamber for 3 min to measure the freezing level in the altered context. A tone (75 dB, 4 kHz) was delivered for 30 s to perform the cue test.

The behavior of the mice was recorded and analyzed with the Video Freeze software. Freezing was defined as the complete absence of movement for at least 0.5 s. On the conditioning day, the freezing percentages were calculated for 30 s during each tone/laser stimulus. For the contextual test, the freezing percentages were calculated for 3 min. For the cue test, the freezing percentages were calculated for 30 s during tone.

## Auditory fear conditioning test

On day 1, mice were habituated to the fear conditioning chambers. On day 2, mice were conditioned by seven trials of sound tone (75 dB, 4 kHz, 30 s) co-terminated with footshock (0.6 mA, 2 s) averagely separated by 90 s. Laser (589 nm, 10 mW) was delivered 1 s before the footshock and lasted for 4 s at each trial. On day 3, mice were placed back into the original training chamber for 3 min to perform the contextual test, and the laser was delivered during the second minute. After 2–3 hr, the mice were placed into a modified chamber to perform the cue test. Three tones were given averagely separated by 90 s. The laser was delivered during the second tone.

The behavior of the mice was recorded and analyzed with the Video Freeze software. The freezing percentages of the 27 s tone before laser (to avoid the influence of laser) for each trial were summarized to indicate fear memory acquisition in the conditioning test. For the contextual test, the freezing percentages were calculated for every minute. For the cue test, the freezing percentages were calculated for 30 s during tone.

## Freezing behavior

For analyses of freezing behavior induced by pharmacogenetic activation of PBN-PVT neurons, we injected CNO and recorded the mouse behavior using the Video Freeze fear conditioning system 30 min later.

The Video Freeze fear conditioning system (MED Associates, MED-VFC-OPTO-USB-M) was also used to assess the effect of optogenetic inhibition of PBN-PVT projection and the PBN-PVT neurons on the fear-like behavior induced by footshock. After free exploration of the chamber for 2 min, 15 times footshocks (0.6 mA, 1 s) were delivered within 10 min with a constant 589 nm laser (10 mW). The freezing percentages during 10 min were analyzed.

The Video Freeze fear conditioning system was also used to assess the effect of optogenetic inhibition of the PBN-PVT neurons on the fear-like behavior induced by 2-MT. 10 mL 2-MT (1:1000) dissolved in the ddH$_2$O was soaked into the cotton ball on the bottom of the training box. A constant laser (589 nm, 10 mW) was delivered during the tests.

## Tail suspension test

Mice were individually suspended by an adhesive tape placed roughly 2 cm from the tip of the tail and videotaped for 6 min. Mice were considered immobile without initiated movements, and the immobility time was scored in the last 3 min by an observer unknown of the treatments.

## Forced swimming test

Mice were individually placed for 6 min in clear cylinders (45 cm height, 20 cm internal diameter) containing freshwater (25°C, 15 cm depth). The swimming activity was videotaped, and immobility

time in the last 3 min was counted manually by an investigator unaware of animal grouping. The mice were considered immobile when they stopped swimming/struggling or only slightly moved to keep the nose above the surface.

## Von Frey test

The von Frey test was used to assess the mechanical sensitivity (*Mu et al., 2017*). The mice were acclimated to the observation chambers for 2 days (2 hr for each day) before the test. A series of von Frey hairs with logarithmically incrementing stiffness (0.16–2.0 g) were used to stimulate the mouse hind paw perpendicularly. The 50% paw withdrawal threshold was determined using the up-down method.

## Hargreaves test

Hargreaves tests were performed as described previously (*Mu et al., 2017*). Mice were placed in an individual plexiglass box with a glass floor. A radiant heat beam was exposed directly to the hind paw until the paw was withdrawn. The trials were repeated three times with an interval of at least 15 min. To avoid potential damage, the test was executed with a 20 s cutoff time.

## Formalin test

In the formalin test, the mice received an intraplantar injection of formalin (5%, 20 µL/mouse) and were placed into a plexiglass box (width: 10 cm; length: 10 cm; height: 15 cm) individually to record the pain-related licking behaviors for 1 hr. All videos were analyzed by trained investigators blinded to the experimental treatment of the animals.

## Rotarod test

Mice were trained twice on a rotarod apparatus (MED Associates) with a rod accelerated 5–20 revolutions per minute (rpm) for 5 min before the experimental day. On the second day, each mouse underwent three trials with a rod programmed to accelerate from 0 to 40 rpm over 300 s, then the average rpm at the point of falling was recorded.

## Fiber photometry

In vivo fiber photometry experiments were performed as previously described (*Zhu et al., 2020*). After 2 weeks of virus expression, the mice were gently handled to be familiar with the calcium signal recording experiments (Thinker-Biotech). A signal (for synchronization) was manually tagged with the shock and air puff to evaluate the activity of PVT neurons. The calcium transient was recorded at 50 Hz. The fluorescence values change ($\Delta F/F$) was calculated from the formula of $(F−F_0)/F_0$, where the $F_0$ represents the median of the fluorescence values in the baseline period ($−1$ to $−0.5$ s relative to the stimulation onset). To precisely quantify the change of the fluorescence values across the shock or air puff stimulation, we defined 0.5–1.0 s after the onset as the post-stimulus period.

## Optoelectrode recording and analysis

The homemade optoelectrode consisted of an optic fiber (200 mm in diameter) glued to 16 individually insulated nichrome wires (35 µm internal diameter, 300–900 Kohm impedance, Stablohm 675, California Fine Wire). The 16 microwire arrays were arranged in a 4 + 4 + 4 + 4 pattern and soldered to an 18-pin connector (Mil-Max). Three weeks after virus injection, the optoelectrode was implanted to the PVT nucleus (AP $−1.46$ mm, ML 0 mm, DV $−2.90$ mm). After 1 week of recovery, two trials were performed continuously. Trial 1 contained 10 sweeps of 2 s laser pulse trains (473 nm, 5 ms, 20 Hz, 8 mW). The interval of sweeps was 60 s. Trial 2 contained 20 sweeps of 2 s footshock (0.5 mA). The interval of sweeps was 60 s. In the even time sweeps (2, 4, 6, 8,10, 12, 14, 16, 18, 20), 2 s laser pulse trains were delivered spontaneously with the 2 s footshock. Neuronal signals were recorded using a Zeus system (Zeus, Bio-Signal Technologies, McKinney, TX), and spike signals were filtered online at 300 Hz. At the end of the experiment, all animals were perfused to confirm the optical fiber sites. Only the data of animals with correct optical fiber sites and virus expression regions were analyzed.

The spikes were sorted by the valley-seeking method with Offline Sorter software (Plexon, USA) and analyzed with NeuroExplorer (Nex Technologies, Boston, MA). Firing rates of the neurons and timestamps were exported for further analysis using customized scripts in MATLAB. The Kolmogorov–Smirnov (K-S) test was used to compare the spike firing rate of PVT during 2 s baseline (before

stimulus) and 2 s after each stimulus. $p < 0.001$ indicates statistical significance. The code generated during this study is available on github (*Xiang, 2021a*, copy archived at swh:1:rev:ce467c67c1c21f-424c92a6e189c7cd96ea938e89; *Xiang, 2021b*). Z-score normalization maps were constructed from normalized firing rates.

To calculate the latency of response of PVT neurons to laser and shock activation, we extracted the data in small bins of 2 ms. The onset of the response was calculated as the first of at least five consecutive bins higher than 20% above the baseline.

## Quantification of the fiber intensity

For quantification of fluorescence of $PVT_{PBN}$ efferents, the downstream targets of $PVT_{PBN}$ neurons were imaged using the identical character, and the mean fluorescence value in each ROI (400 × 400 pixels) of each brain region was analyzed using Fiji. The fiber intensity was calculated as the fluorescence value of each brain region divided by that of the NAc. All data came from at least three different mice and are presented as mean ± SEM.

## Analysis

Statistical detection methods include unpaired Student's *t*-test, paired Student's *t*-test, one-way ANOVA with Bonferroni's correction for multiple comparisons, two-way ANOVA with Bonferroni's correction for multiple comparisons. A value of $p < 0.05$ is considered statistically significant. All data are represented as mean ± SEM.

## Acknowledgements

We thank Dr. Yan-Gang Sun for comments on the manuscript and for providing Vglut2-ires-Cre mice. We thank Dr. Hua-Tai Xu for providing *Rosa26-tdTomato* mice. We thank all the lab members of DM for their helpful discussion. This work was supported by the National Natural Science Foundation of China (No. 31900717, 31571086), the Shanghai Sailing Program (19YF1438700 to DM), and the Young Elite Scientists Sponsorship Program of China Association for Science and Technology (2019QNRC001 to DM), the Shanghai Natural Science Foundation (21ZR1468600 to LZ).

## Additional information

### Funding

| Funder | Grant reference number | Author |
|---|---|---|
| National Natural Science Foundation of China | 31900717 | Di Mu |
| China Association for Science and Technology | 2019QNRC001 | Di Mu |
| Shanghai Association for Science and Technology | 19YF1438700 | Di Mu |
| National Natural Science Foundation of China | 31571086 | Ling Zhang |
| Shanghai Natural Science Foundation | 21ZR1468600 | Ling Zhang |

The funders had no role in study design, data collection and interpretation, or the decision to submit the work for publication.

### Author contributions

Ya-Bing Zhu, Data curation, Formal analysis, Investigation, Methodology, Software, Writing - original draft; Yan Wang, Rui Zhang, Formal analysis, Investigation, Validation; Xiao-Xiao Hua, Ling Xu, Ming-Zhe Liu, Jin-Bao Li, Investigation; Peng-Fei Liu, Investigation, Validation; Ling Zhang, Funding acquisition, Project administration; Di Mu, Conceptualization, Funding acquisition, Project administration, Resources, Supervision, Writing - original draft, Writing - review and editing

## Author ORCIDs
Ling Zhang http://orcid.org/0000-0002-8308-6553
Di Mu http://orcid.org/0000-0003-1209-9311

## Ethics
All animal experiment procedures were approved by the Animal Care and Use Committee of Shanghai General Hospital (2019AW008).

## Decision letter and Author response
Decision letter https://doi.org/10.7554/eLife.68372.sa1
Author response https://doi.org/10.7554/eLife.68372.sa2

## Additional files

### Supplementary files
• Transparent reporting form

### Data availability
All data generated or analysed during this study are included in the manuscript and supporting file. The behavioral data and imaging analysis results have been made available on Dryad Digital Repository. All MATLAB code has been deposited on Github (copy archived at swh:1:rev:ce467c67c1c21f-424c92a6e189c7cd96ea938e89) and is publicly available.

The following dataset was generated:

| Author(s) | Year | Dataset title | Dataset URL | Database and Identifier |
|---|---|---|---|---|
| Mu D | 2022 | PBN-PVT projections modulate negative affective states in mice | http://dx.doi.org/10.5061/dryad.1rn8pk0w4 | Dryad Digital Repository, 10.5061/dryad.1rn8pk0w4 |

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
