## [Editor Report]

This study will interest neuroscientists, in particular those interested in the neural circuits that support emotional processing. Using modern neuroscience techniques, the authors demonstrate that anatomical projections from the parabrachial nucleus to the paraventricular nucleus thalamus contribute to aversive states like fear and anxiety. Overall, the study offers important details of a previously uncharacterized brain circuit.

---

## [Decision Letter]

**Decision letter after peer review:**

Thank you for submitting your article "PBN-PVT projection modulates negative emotions in mice" for consideration by *eLife*. Your article has been reviewed by 3 peer reviewers, including Mario Penzo as the Reviewing Editor and Reviewer #1, and the evaluation has been overseen by Andrew King as the Senior Editor. The following individual involved in review of your submission has agreed to reveal their identity: Fabricio do Monte (Reviewer #2).

Essential revisions:

While individual assessments and recommendations from each of the reviewer is included below, here we provide you with a brief list of items that we collectively consider to be essential revisions that must be addressed in order for the manuscript to be considered further for publication at *eLife*.

1) Revise the writing and overall framing of the study. The authors should avoid anthropomorphizing rodents by using terms typically associated with human's subjective experience of emotions like sadness and misery.

2) The authors should provide a more detailed discussion and present the temporal profiles of the behaviors elicited by photo stimulating PBN-PVT projections (see comments from Reviewer # 2).

3) While the authors showed that VGLUT2 expressing neurons of the PBN project to the PVT, we ask that the authors perform a more complete characterization of the molecular identity of PVT-projecting PBN neurons. The authors are not expected to identify new markers, but rather use known PBN markers as reference. The objective should be to determine what fraction of PBN projections to the PVT arise from VGLUT2^+^ neurons vs other PBN markers.

4) The authors should demonstrate that PBN input shapes PVT neuronal responses to aversive stimuli by expanding on their findings presented in Figure 5.

*Reviewer #1 (Recommendations for the authors):*

1) I recommend that the authors reconsider their overall framing of the term emotion and its evaluation in animal models. While in general the writing of the manuscript could be improved, I found the use of subjective terms of emotion like sadness and misery particularly problematic. Framing of the study within the constraints of aversion may be more appropriate here.

2) In Figure 1 it appears that the density of PBN projections is higher in the middle and posterior PVT (pPVT). The authors should better highlight this observation in the text, especially considering that the notion of the pPVT being a particularly aversive region of the PVT is now supported by various studies. The overall framing of the PBN-PVT pathway as involved in aversion fits well with the anatomical distribution of these projections.

3) Justification for the use of different genetic strategies to target the PBN-PVT pathway is required. When using the constitutively expensed ChR2 in the PBN to demonstrate glutamatergic projections to the PVT, please show whether there is contamination from adjacent brain stem structures like the LC and DRN, which project to the PVT and are known to contain glutamatergic neurons (vglut1 and vglut3, respectively). In figure 4 why were Vglut2-cre mice not used and PBN terminals in the PVT inhibited as in Figure 2?

4) In the retrograde labeling experiment (Figure 1) please show what fraction of retrogradely label cells are VGLUT2^+^. You should also ensure that a similar population of PBN-PVT neurons has been assessed throughout the study.

5) Please reassess the statistical tests used, as outlined in the Public Review section.

6) Please consider whether associative memory (e.g. fear memory retrieval) is impacted by manipulations of the PBN-PVT pathway.

7) Please provide better evidence for whether increased activity of PVT neurons is mediated by input from the PBN.

*Reviewer #2 (Recommendations for the authors):*

1. This manuscript has numerous grammatical issues and typos. It would benefit enormously if the text is carefully proofread. It also misuses terminologies and concepts in several cases, including the use of the word "affection" presumably to refer to "affective states", or the use of the word "rescue" to describe the reduction of "fear-like behaviors". These issues should be corrected.

2. The authors frequently refer to Chiang et al. 2020 (PMID: 32289251) paper to support their observation that PBN-PVT projections are exclusively glutamatergic. However, the same paper showed that the PBN also sends a smaller number of GABAergic fibers to PVT. While the authors showed only EPSCs in PVT neurons when optogenetically activating PBN inputs, it is not clear whether the recordings were also performed after clamping the voltage at 0mV to allow the detection of possible IPSCs after stimulation of PBN fibers. Alternatively, the authors could use immunohistochemistry to identify the cell-type specificity of PVT-projecting neurons in PBN and reconcile their findings with the current literature.

3. Similarly, Chiang et al. 2020 described the role of PBN-CeA/BNST projections in regulating aversion. Although the authors showed that PVT-projecting neurons in PBN do not project to CeA, it would be very informative for the field if the authors could also provide the data for other brain regions to which PVT-projecting neurons also send collateral, especially those regions that are important for anxiety- and fear- like behaviors. This would help the authors to exclude (or include) the possible contribution of collateral projections to the reported behaviors.

4. For the PBN-PVT photoactivation experiments in Figure 2, a deeper discussion about the lack of effects on aversive memory formation (RTPA and fear conditioning) is missing. The authors should compare their results with prior studies in the literature. For example, activation of PVT-CeA projections induces place aversion and the effect persists in the next day in the absence of illumination (Do Monte et al. 2017, PMID: 28426970). Similarly, LTD-like stimulation of PVT-CeA projections or optogenetic/chemogenetic inhibition of the same circuit induces a persistent attenuation of fear responses (Chen and Bi, 2019, PMID: 30406427; Do Monte et al. 2015, PMID: 25600268; Penzo et al. 2015, PMID: 25600269)

5. In Figure 4, the authors should justify why they have inhibited the soma of PVT-projecting PBN neurons instead of axonal terminals in the PVT. They should also make it very clear in their schematic drawing by adding an optical fiber on the top of the PVT, similar to what they have done in other figures.

6. During optogenetic manipulations, the authors used a very long period of laser illumination (5 uninterrupted minutes of pulses for photoactivation and 10 minutes of constant laser for photoinhibition). These laser durations are much longer than the traditional intervals used for optogenetic manipulations in rodents. An explanation of the rationale for using such long intervals is needed. Also, regarding the photoactivation experiment (illustrated in Supplementary Video 1), it is very clear that the animals exhibited a wide range of different behaviors along the 5-min period of stimulation. These behaviors varied from no effect to increased speed of locomotion, to jumping episodes, to freezing responses by the end of the stimulation, thereby making it hard to define the correct function of this pathway in behavioral regulation. A breakdown of the behaviors by time and a deeper discussion of the different responses would probably help the readers to better understand the results.

7. The authors should provide a Cre-negative control for Figure 5G-N, given that tdTomato Ai14 mice can have baseline levels of tdTomato expression even in the absence of Cre. This is highly important because the transneuronal transduction of AAV1-Cre is generally limited and has never been tested in PVT.

*Reviewer #3 (Recommendations for the authors):*

1. It has been shown that PBN-VMH/PAG drive escape behaviors, whereas PBN-BNST/CEA generate aversive memory. The authors need to examine the relationship between PVT-projection and VMH/BNST-projections.

2. The authors need to examine the cellular identity of PVT-projecting PBN neurons, whether it originate from known PBN neuronal subtypes such as CGRP, Tac1, Pdyn, Nts et al.

3. As authors mentioned in discussion part, a recent study revealed that the PBN neurons project to ILN, which is relatively closed to the PVT. The authors need to carefully examine their manipulation experiments especially optogenetic stimulation. They need to make sure that the effect they observed was not due to the activation of PBN-ILN pathway. Since activation of PBN-ILN pathway induced pain-related behavior which could be very similar to negative emotions.

4. For the cue-dependent optogenetics conditioning test: activation of the PBN-PVT projection induces instant aversion and freezing but not drives associative fear learning. Could it be simply because the conditioning (6 times 30 seconds light stimulation) was too weak? The authors should try a prolong conditioning protocol that resemble drug CPP such as 30 min light stimulation at paired side.

5. Optogenetics and Pharmacogenetics are nouns while optogenetic and pharmacogenetic are adjectives. There are many places in the manuscript where the noun was misused.

[Editors' note: further revisions were suggested prior to acceptance, as described below.]

Thank you for resubmitting your work entitled "PBN-PVT projections modulate negative affective states in mice" for further consideration by *eLife*. Your revised article has been evaluated by Andrew King (Senior Editor) and a Reviewing Editor.

The manuscript has been improved but there are some remaining issues that need to be addressed, as outlined below:

The reviewers collectively agree that, in the present revision, the authors have addressed the most relevant criticism raised following the initial submission. As such, we consider that the manuscript is a strong candidate for publication in *eLife*. However, before reaching a final decision on the manuscript, we recommend that the authors address the remaining recommendations, particularly from Reviewer #2. No additional experiments are requested at this time. Revisions are strictly for clarity purposes.

*Reviewer #1 (Recommendations for the authors):*

In their revised manuscript, the authors have adequately addressed the previous concerns and recommendations raised by this reviewer. Importantly, doing so has improved the overall quality of the manuscript.

The manuscript could benefit from additional copy editing to avoid statements such as "PVT neurons innervated by PBN innervation" (Line 39).

*Reviewer #2 (Recommendations for the authors):*

In this revised version of the manuscript, the authors did a good job fully addressing all the concerns raised during the first submission. They have now re-framed the manuscript with careful interpretation and discussion of the results. They also carried out new experiments to address some important questions that were unclear in the first version of the manuscript. As a result, the clarity and quality of the manuscript has significantly improved compared to the previous version.

*Reviewer #3 (Recommendations for the authors):*

The authors have carefully addressed all my questions and I appreciate their efforts in revising manuscript. I have no further comments.

---

## [Author Response]

Essential revisions:While individual assessments and recommendations from each of the reviewer is included below, here we provide you with a brief list of items that we collectively consider to be essential revisions that must be addressed in order for the manuscript to be considered further for publication at eLife.1) Revise the writing and overall framing of the study. The authors should avoid anthropomorphizing rodents by using terms typically associated with human's subjective experience of emotions like sadness and misery.

We have revised the text to avoid using subjective terms of emotions. We used terms including anxiety-like, depression-like, fear-like behaviors, freezing, aversion-like behaviors. We replaced the terms “emotions” “affections” with “affective states”. We also replaced the terms “rescue” “relieve” with “reduce”.

2) The authors should provide a more detailed discussion and present the temporal profiles of the behaviors elicited by photo stimulating PBN-PVT projections (see comments from Reviewer # 2).

We have now provided a detailed analysis (one-minute time window) of optogenetic activation of PBN-PVT projections in OFT experiments. We further divided the laser ON period (5−10 minutes) into five one-minute periods and analyzed the velocity, non-moving time, center time, moving distance, and jumping behaviors. We found that the velocity and non-moving time were increased, and the center time was decreased in the ChR2 mice during most periods. Furthermore, we observed that the distance and jumping behaviors were increased mainly in the first one-minute period in ChR2 mice. This detailed analysis indicated that optogenetic activation induced brief and robust running, jumping behaviors, and persistent anxiety-like behaviors, such as less time spent in the center. These new results have been included in *Figure 2−figure supplement 2* and were described in the text. Please see Page 7 Line 179 to Line 189. We also discussed this on Page 14 Line 396 to Line 403.

3) While the authors showed that VGLUT2 expressing neurons of the PBN project to the PVT, we ask that the authors perform a more complete characterization of the molecular identity of PVT-projecting PBN neurons. The authors are not expected to identify new markers, but rather use known PBN markers as reference. The objective should be to determine what fraction of PBN projections to the PVT arise from VGLUT2^+^ neurons vs other PBN markers.

To examine the molecular identity of PVT-projecting PBN neurons, we have now performed the RNAscope experiments detecting *VgluT2, Tac1, Tacr1, Pdyn* mRNA, and fluorescent immunostaining detecting CGRP protein in the PBN. We injected the retroAAV-Cre virus into the PVT nucleus on *Rosa26-tdTomato* mice and found that tdTomato^+^ neurons were located in the PBN. We found that about 94.4% of tdTomato^+^ neurons express *VgluT2* mRNA*.* These results indicate that the majority of PVT-projecting PBN neurons are glutamatergic. These new results have been included in *Figure 1R−U* and were described in the text. Please see Page 5 Line 129 to Line 132.

We further determined the identity of PVT-projecting PBN neurons as suggested by Reviewer #3. We found that tdTomato^+^ neurons were only partially co-labeled with *Tacr1*, *Tac1,* or *Pdyn* mRNA, but not with CGRP. These new results have been included in *Figure 1−figure supplement 1* and were described in the text. Please see Page 5 Line 132 to Line 140.

4) The authors should demonstrate that PBN input shapes PVT neuronal responses to aversive stimuli by expanding on their findings presented in Figure 5.

We have now performed the optoeletrode experiments to examine the effects of activation of PBN-PVT projections on PVT neuronal responses to footshock.

We injected AAV-DIO-ChR2 virus into the PBN and implanted the optoelectrode into the PVT of *VgluT2-ires-Cre* mice. We first recorded the spiking signals in response to 10 sweeps of 2-second laser pulse trains. Then we recorded the spiking signals in response to 20 sweeps of 2-second footshock without laser in the odd number sweeps or with laser in the even number sweeps. We found that there was also a broad overlap between laser-activated and footshock-activated neurons. Next, we analyzed the firing rates of PVT neurons during footshock with laser sweeps and footshock without laser sweeps. We found that the footshock stimulus with laser activated 30 of 40 neurons and increased the overall firing rates of 40 neurons compared with the footshock without laser result. These results indicated that activation of PBN-PVT projections could enhance PVT neuronal responses to aversive stimulation. These new results have been included in *Figure 6* and were described in the text. Please see Page 11 Line 302 to Line 319.

Reviewer #1 (Recommendations for the authors):1) I recommend that the authors reconsider their overall framing of the term emotion and its evaluation in animal models. While in general the writing of the manuscript could be improved, I found the use of subjective terms of emotion like sadness and misery particularly problematic. Framing of the study within the constraints of aversion may be more appropriate here.

We agree with the reviewer. Now we have reframed this manuscript. We have revised the text to avoid using subjective terms of emotions. We used terms including anxiety-like, depression-like, fear-like behaviors, freezing, aversion-like behaviors. We replaced the terms “emotions” “affections” with “affective states”. We also replaced the terms “rescue” “relieve” with “reduce”.

2) In Figure 1 it appears that the density of PBN projections is higher in the middle and posterior PVT (pPVT). The authors should better highlight this observation in the text, especially considering that the notion of the pPVT being a particularly aversive region of the PVT is now supported by various studies. The overall framing of the PBN-PVT pathway as involved in aversion fits well with the anatomical distribution of these projections.

We injected AAV2/8-EF1α-DIO-EGFP into PBN of *VgluT2-ires-Cre* mice. It is worth noting that the density of EGFP^+^ fibers was higher in the middle and posterior PVT (*Figure 1−figure supplement 2E−H*), considering the notion that the posterior PVT is particularly sensitive to aversion (*Gao et al.,* 2020). Please see Page 5 Line 145 to Page 6 Line 148. We also discussed this on Page 14 Line 387 to Line 391.

We also checked the recording sites of the PVT neurons in the PBN-PVT connectivity experiment. Some neurons should be marked as posterior PVT neurons rather than medial PVT, and we have revised the result. Please see *Figure 1D−F*, Page 4 Line 108 to Line 112.

3) Justification for the use of different genetic strategies to target the PBN-PVT pathway is required. When using the constitutively expensed ChR2 in the PBN to demonstrate glutamatergic projections to the PVT, please show whether there is contamination from adjacent brain stem structures like the LC and DRN, which project to the PVT and are known to contain glutamatergic neurons (vglut1 and vglut3, respectively). In figure 4 why were Vglut2-cre mice not used and PBN terminals in the PVT inhibited as in Figure 2?

We agree with the reviewer. Now we have reframed this manuscript. We first presented the slice recording results from wild-type mice (*Figure 1*). We found that light-induced EPSCs in 34 of 52 neurons and light-induced IPSCs in 4 of 52 neurons. Please see Page 5 Line 119 to Line 121.

We carefully examined the ChR2 virus infection area. The raphe nucleus was not infected. Some neurons in the LC were infected with the AAV-hSyn-ChR2 virus. We agreed with the reviewer that there could be potential contamination from the LC, which releases dopamine and norepinephrine to the PVT by LC-PVT projection. We have discussed this on Page 13 Line 375 to Line 380.

We performed tdTomato staining with *VgluT2* mRNA in situ hybridization and found that about 94.4% of tdTomato^+^ neurons express *VgluT2* mRNA*.* These new results indicate that the majority of PVT-projecting PBN neurons are glutamatergic. Please see *Figure 1R−U* and Page 5 Line 129 to Line 132.

Then we used *VgluT2-ires-Cre* mice to perform tracing (*Figure1−figure supplement 2*) and behavioral tests (optogenetic activation in *Figure 2*, optogenetic inhibition in *Figure 4*). We also performed the pharmacogenetic activation of PVT-projecting PBN neurons on wild-type mice (*Figure 3*). We observed that pharmacogenetic activation of the PVT-projecting PBN neurons reduced the center duration in the OFT, similar to the optogenetic activation OFT result. We also observed that pharmacogenetic activation of the PVT-projecting PBN neurons induced freezing behaviors. Our pharmacogenetic activation experiment supported the hypothesis that PBN-PVT projections modulate negative affective states.

Now we have performed the optogenetic inhibition of the PBN-PVT projections using *VgluT2-ires-Cre* mice. We found that inhibition of PBN-PVT projections reduces 2-MT-induced aversion-like behaviors and footshock-induced freezing behaviors. These new results have been included in *Figure 4*, *Figure 4−figure supplement 1 and 2*, and were described in the text. Please see Page 9 Line 254 to Page 10 Line 274.

4) In the retrograde labeling experiment (Figure 1) please show what fraction of retrogradely label cells are VGLUT2^+^. You should also ensure that a similar population of PBN-PVT neurons has been assessed throughout the study.

We performed tdTomato staining with *VgluT2* mRNA in situ hybridization and found that about 94.4% of tdTomato^+^ neurons express *VgluT2* mRNA*.* These results indicate that the majority of PVT-projecting PBN neurons are glutamatergic. These new results have been included in *Figure 1R−U* and described in the text. Please see Page 5 Line 129 to Line 132.

5) Please reassess the statistical tests used, as outlined in the Public Review section.

We have revised the statistics (Unpaired student’s *t*-test) and calculated the percentage of freezing behaviors in two groups during 10-minute experiment, which matched the constant optogenetic inhibition. Please see *Figure 4K* and related figure legend. Similar changes have been made in the *Figure 4−figure supplement 3K.*

6) Please consider whether associative memory (e.g. fear memory retrieval) is impacted by manipulations of the PBN-PVT pathway.

We have now performed several experiments to examine the effects of the PBN-PVT projections on aversion formation and memory retrieval.

We first performed a prolonged conditioned place aversion that mimics drug-induced place aversion. And we found that optogenetic activation of PBN-PVT projections did not induce aversion in the postconditioning test on Day 4. These new results have been included in *Figure 2−figure supplement 2H−I* and described in the text. Please see Page 7 Line 196 to Line 199.

Then, we performed the classical auditory fear conditioning test and found that optogenetic inhibition of PBN-PVT projections during footshock in the conditioning period did not affect freezing levels in contextual test or cue test (Laser OFF trials). And inhibition of the PBN-PVT projection during contextual test or cue test (Laser On trials) did not affect freezing levels either. These data suggest that PBN-PVT projections are not crucial for associative fear memory formation or retrieval. These new results have been included in *Figure 4−figure supplement 2* and described in the text. Please see Page 10 Line 268 to Line 274. We also discussed this on Page 15 Line 430 to Page 16 Line 473.

7) Please provide better evidence for whether increased activity of PVT neurons is mediated by input from the PBN.

We have now performed the dual Fos staining experiment and the optoeletrode experiment.

In the dual Fos staining experiment, we found that there was a broad overlap between optogenetic stimulation-activated neurons (expressing the Fos protein) and footshock-activated neurons (expressing the *fos* mRNA) (*Figure 6−figure supplement 1B−E*).

In optoelectrode experiment, there was also a broad overlap between laser-activated and footshock-activated neurons. This result was consistent with the dual Fos staining result, suggesting that PVT_PBN_ neurons were activated by aversive stimulation. Next, we analyzed the firing rates of PVT neurons during footshock with laser sweeps and footshock without laser sweeps. We found that the footshock stimulus with laser activated 30 of 40 neurons and increased the overall firing rates of 40 neurons compared with the footshock without laser result (*Figure 6I*). These results indicated that activation of PBN-PVT projections could enhance PVT neuronal responses to aversive stimulation.

These new results have been included in *Figure 6*, *Figure 6−figure supplement 1*, and described in the text. Please see Page 10 Line 295 to Page 11 Line 319. We also discussed these results on Page 15 Line 422 to Line 429.

Reviewer #2 (Recommendations for the authors):1. This manuscript has numerous grammatical issues and typos. It would benefit enormously if the text is carefully proofread. It also misuses terminologies and concepts in several cases, including the use of the word “affection” presumably to refer to “affective states”, or the use of the word “rescue” to describe the reduction of “fear-like behaviors”. These issues should be corrected.

We have revised the text to avoid using subjective terms of emotions. We used terms including anxiety-like, depression-like, fear-like behaviors, freezing, aversion-like behaviors. We replaced the terms “emotions” “affections” with “affective states”. We also replaced the terms “rescue” “relieve” with “reduce”.

2. The authors frequently refer to Chiang et al. 2020 (PMID: 32289251) paper to support their observation that PBN-PVT projections are exclusively glutamatergic. However, the same paper showed that the PBN also sends a smaller number of GABAergic fibers to PVT. While the authors showed only EPSCs in PVT neurons when optogenetically activating PBN inputs, it is not clear whether the recordings were also performed after clamping the voltage at 0mV to allow the detection of possible IPSCs after stimulation of PBN fibers. Alternatively, the authors could use immunohistochemistry to identify the cell-type specificity of PVT-projecting neurons in PBN and reconcile their findings with the current literature.

Now we have revised the overall framing of the study. We first presented the slice recording results from wild-type mice (*Figure 1*). We found that light-induced EPSCs in 34 of 52 neurons and light-induced IPSCs in 4 of 52 neurons. Please see Page 5 Line 119 to Line 121 in the result section. The small portion of inhibitory connections is consistent with the previous anatomical result. We also discussed this on Page 13 Line 380 to Line 383.

We also detected several molecular markers (*Vglut2, Tac1, Tacr1, Pdyn,* CGRP) in the PVT-projecting PBN neurons on Ai9 mice and found that about 94% of tdTomato^+^ neurons expressed *VgluT2* mRNA. These new results have been included in *Figure 1* and *Figure 1−figure supplement 1*. Please see Page 5 Line 129 to Line 140.

3. Similarly, Chiang et al. 2020 described the role of PBN-CeA/BNST projections in regulating aversion. Although the authors showed that PVT-projecting neurons in PBN do not project to CeA, it would be very informative for the field if the authors could also provide the data for other brain regions to which PVT-projecting neurons also send collateral, especially those regions that are important for anxiety- and fear- like behaviors. This would help the authors to exclude (or include) the possible contribution of collateral projections to the reported behaviors.

We have now provided the distribution pattern of collateral projections from PVT-projecting PBN neurons. The collateral projections were found in BNST, LH, PVN, PAG but not CeA or VMH. These new results have been included in *Figure 1−figure supplement 3* and described in the text. Please see Page 6 Line 148 to Line 151.

Recently, Chiang *et al.* found two major efferent pathways from the lateral PBN: one originating from the dorsal division of lateral PBN collateralizes the VMH and PAG and a second arising from the external lateral division of lateral PBN that collateralize to the BNST and CeA. They suggested that activation of the first pathway generates the aversive memory and activation of the second one drives escape behaviors *(Chiang et al., 2020)*.

In our study, the results indicate that the PBN-PVT pathway arises from both dorsal and external lateral divisions of lateral PBN and collateralizes the BNST, LH, PVN, and PAG but not the CeA or VMH. According to the location of originated PBN neurons and collateral projection pattern, we speculated that the PBN-PVT efferent pathway is different from both PBN-VMH/PAG pathway and PBN-BNST/CeA pathway. We discussed this on Page 15 Line 444 to Page16 Line 457.

4. For the PBN-PVT photoactivation experiments in Figure 2, a deeper discussion about the lack of effects on aversive memory formation (RTPA and fear conditioning) is missing. The authors should compare their results with prior studies in the literature. For example, activation of PVT-CeA projections induces place aversion and the effect persists in the next day in the absence of illumination (Do Monte et al. 2017, PMID: 28426970). Similarly, LTD-like stimulation of PVT-CeA projections or optogenetic/chemogenetic inhibition of the same circuit induces a persistent attenuation of fear responses (Chen and Bi, 2019, PMID: 30406427; Do Monte et al. 2015, PMID: 25600268; Penzo et al. 2015, PMID: 25600269)

We have now performed two new experiments: the prolonged conditioned place aversion test and the classical auditory fear conditioning test. The results suggested that PBN-PVT projections are not crucial for associative aversive memory formation. We also discussed this on Page 15 Line 430 to Page 16 Line 473.

We first performed a prolonged conditioned place aversion that mimics drug-induced place aversion. And we found that optogenetic activation of PBN-PVT projections did not induce aversion in the postconditioning test on Day 4. These new results have been included in *Figure 2−figure supplement 2* and described in the text. Please see Page 7 Line 196 to Line 199.

Then, we performed the classical auditory fear conditioning test and found that optogenetic inhibition of PBN-PVT projections during footshock in the conditioning period did not affect freezing levels in contextual test or cue test (Laser OFF trials). And inhibition of PBN-PVT projections during contextual test or cue test (Laser On trials) did not affect freezing levels either. These data suggest that PBN-PVT projections are not crucial for associative fear memory formation or retrieval. These new results have been included in *Figure 4−figure supplement 2* and described in the text. Please see Page 10 Line 268 to Page Line 274.

5. In Figure 4, the authors should justify why they have inhibited the soma of PVT-projecting PBN neurons instead of axonal terminals in the PVT. They should also make it very clear in their schematic drawing by adding an optical fiber on the top of the PVT, similar to what they have done in other figures.

we have now performed the optogenetic inhibition of the PBN-PVT projection using *VgluT2-ires-Cre* mice. These new results have been included in *Figure 4* and described in the text. Please see Page 9 Line 254 to Page 10 Line 274.

6. During optogenetic manipulations, the authors used a very long period of laser illumination (5 uninterrupted minutes of pulses for photoactivation and 10 minutes of constant laser for photoinhibition). These laser durations are much longer than the traditional intervals used for optogenetic manipulations in rodents. An explanation of the rationale for using such long intervals is needed. Also, regarding the photoactivation experiment (illustrated in Supplementary Video 1), it is very clear that the animals exhibited a wide range of different behaviors along the 5-min period of stimulation. These behaviors varied from no effect to increased speed of locomotion, to jumping episodes, to freezing responses by the end of the stimulation, thereby making it hard to define the correct function of this pathway in behavioral regulation. A breakdown of the behaviors by time and a deeper discussion of the different responses would probably help the readers to better understand the results.

There are several studies using relatively short-term optogenetic activation protocols in fear conditioning experiments (*Do-Monte et al.,* 2015; *Gao et al.,* 2020). And there are also studies using relatively long-term optogenetic manipulation protocols in OFT, TST, sucrose preference, and pain-related experiments. For example, Zhou *et al.* used 30 minutes constant yellow laser (594 nm, 5-8 mW) and 5-minute pulse of blue light (473 nm, 1-3 mW, 15 ms pulses, 20 Hz) (*Zhou et al.,* 2019). Another group used 5 min OFF/5 min ON/5 min OFF protocol to manipulate PBN neurons (*Sun et al.,* 2020). They used 473 nm (20 Hz, 5 ms pulse width, 5 mW) for activating ChR2 and used 589 nm (1 Hz, 999-ms pulse width, 10 mW) for activating NpHR.

Compared with these long-term optogenetic studies, our optogenetic protocol for ChR2 (5 minutes) is acceptable, but the protocol for NpHR (10 minutes constant inhibition) is relatively long. The main reason was the effect of inhibiting the PBN-PVT projection was not as significant as the effect of optogenetic activation. We needed to perform relatively long-term behavior tests (10 minutes in the 2-MT CPA aversion test and 10 minutes in the footshock test) to analyze the results. We discussed this on Page 14 Line 413 to Page 15 Line 418.

We have provided a detailed analysis (one-minute time window) of optogenetic activation of PBN-PVT projections in OFT experiments. We further divided the laser ON period (5−10 minutes) into five one-minute periods and analyzed the velocity, non-moving time, center time, travel distance, and jumping behaviors (*Figure 2−figure supplement 2C−G*). We found that the velocity and non-moving time were increased, and the center time was decreased in the ChR2 mice during most periods (*Figure 2−figure supplement 2C−E*). Furthermore, we observed that the distance and jumping behaviors were increased mainly in the first one-minute period in ChR2 mice (*Figure 2−figure supplement 2F−G*). This detailed analysis indicated that optogenetic activation induced brief and robust running, jumping behaviors, and persistent anxiety-like behaviors, such as less time spent in the center. These new results have been included in *Figure 2−figure supplement 2* and described in the text. Please see Page 7 Line 179 to Line 189. We also discussed this on Page 14 Line 396 to Line 403.

7. The authors should provide a Cre-negative control for Figure 5G-N, given that tdTomato Ai14 mice can have baseline levels of tdTomato expression even in the absence of Cre. This is highly important because the transneuronal transduction of AAV1-Cre is generally limited and has never been tested in PVT.

We agree with the reviewer. We used the Ai9 allele (Stock No. 007909). “Both Ai14 and Ai9 may exhibit low levels of tdTomato expression prior to exposure to Cre recombinase – but the tdTomato expression levels after Cre recombination are greater than those baseline levels. As such, it is recommended that researchers include Cre-negative controls to establish the baseline tdTomato levels in their experiments” (from Jax website)*.*

We perfused one Ai9 mouse and carefully checked the baseline tdTomato expression. We found that there were no obvious tdTomato^+^ neurons in the PVT. Please see Figure 1B. We speculated that the baseline level of tdTomato expression was relatively poor, and we could not detect the tdTomato fluorescence signal in the absence of Cre.

Reviewer #3 (Recommendations for the authors):1. It has been shown that PBN-VMH/PAG drive escape behaviors, whereas PBN-BNST/CEA generate aversive memory. The authors need to examine the relationship between PVT-projection and VMH/BNST-projections.

We have provided the distribution pattern of collateral projections from PVT-projecting PBN neurons. The collateral projections were found in BNST, LH, PVN, PAG but not CeA or VMH. These new results have been included in *Figure 1−figure supplement 3* and described in the text. Please see Page 6 Line 148 to Line 151.

Recently, Chiang *et al.* found two major efferent pathways from the lateral PBN: one originating from the dorsal division of lateral PBN collateralizes the VMH and PAG and a second arising from the external lateral division of lateral PBN that collateralize to the BNST and CeA. They suggested that activation of the first pathway generates the aversive memory, and activation of the second one drives escape behaviors *(Chiang et al., 2020)*.

In our study, the results indicate that the PBN-PVT pathway arises from both dorsal and external lateral divisions of lateral PBN and collateralizes the BNST, LH, PVN, and PAG but not the CeA or VMH. According to the location of originated PBN neurons and collateral projection pattern, we speculated that the PBN-PVT efferent pathway is different from both PBN-VMH/PAG pathway and PBN-BNST/CeA pathway. We discussed this on Page 15 Line 444 to Page 16 Line 457.

2. The authors need to examine the cellular identity of PVT-projecting PBN neurons, whether it originates from known PBN neuronal subtypes such as CGRP, Tac1, Pdyn, Nts et al.

To examine the molecular identity of PVT-projecting PBN neurons, we have now performed the RNAscope experiments detecting *VgluT2, Tac1, Tacr1, Pdyn* mRNA, and fluorescent immunostaining detecting CGRP protein in the PBN. We found that about 94.4% of tdTomato^+^ neurons expressed *VgluT2* mRNA. We also found that tdTomato^+^ neurons were only partially co-labeled with *Tacr1* mRNA (38.0%), *Tac1* mRNA (6.4%)*,* or *Pdyn* mRNA (23.0%), but not with CGRP.

These results indicate that the majority of PVT-projecting PBN neurons are glutamatergic neurons and these neurons do not specifically originate from subpopulations labeled by Tacr1, Tac1, Pdyn, or CGRP. These new results have been included in *Figure 1*, *Figure 1−figure supplement 1*, and were described in the text. Please see Page 5 Line 129 to Line 140.

3. As authors mentioned in discussion part, a recent study revealed that the PBN neurons project to ILN, which is relatively closed to the PVT. The authors need to carefully examine their manipulation experiments especially optogenetic stimulation. They need to make sure that the effect they observed was not due to the activation of PBN-ILN pathway. Since activation of PBN-ILN pathway induced pain-related behavior which could be very similar to negative emotions.

We have now provided the virus expression and optic fiber tips locations. Please see *Figure 2−figure supplement 1* (ChR2 experiments) and Figure 4*−figure supplement 1* (NpHR3.0 experiments). The virus was expressed in the PBN, and the optic fibers were above the PVT. These results confirmed that we manipulated the PBN-PVT pathway without affecting the PBN-ILN pathway.

We also performed pharmacogenetic activation of PVT-projecting PBN neurons and found that basal nociceptive thresholds or the formalin-induced pain behaviors were not affected (*Figure 3−figure supplement 2*). We speculated that PBN-PVT pathway might not be involved in nociceptive information processing. We also discussed this on Page 17 Line 494 to Line 502.

4. For the cue-dependent optogenetics conditioning test: activation of the PBN-PVT projection induces instant aversion and freezing but not drives associative fear learning. -- Could it be simply because the conditioning (6 times 30 seconds light stimulation) was too weak? The authors should try a prolong conditioning protocol that resemble drug CPP such as 30 min light stimulation at paired side.

We agree with the reviewer. We have now performed a prolonged conditioned place aversion that mimics drug-induced condition place aversion (*Figure 2−figure supplement 2H−I*), and we found that optogenetic activation of the PBN-PVT projection did not induce aversion in the postconditioning test on Day 4.

These new results have been included in *Figure 2−figure supplement 2H−I*, and were described in the text. Please see Page 7 Line 196 to Line 199. We speculated that PBN-PVT projections might not be crucial for associative fear learning. We also discussed this on Page 15 Line 430 to Page 16 Line 473.

5. Optogenetics and Pharmacogenetics are nouns while optogenetic and pharmacogenetic are adjectives. There are many places in the manuscript where the noun was misused.

We have revised the text.

References

Do-Monte FH, Quinones-Laracuente K, Quirk GJ. (2015). A temporal shift in the circuits mediating retrieval of fear memory. *Nature,* 519:460-463. DOI: https://doi.org/10.1038/nature14030, PMID:25600268

Gao C, Leng Y, Ma J, Rooke V, Rodriguez-Gonzalez S, Ramakrishnan C, Deisseroth K, Penzo MA. (2020). Two genetically, anatomically and functionally distinct cell types segregate across anteroposterior axis of paraventricular thalamus. *Nat Neurosci,* 23:217-228. DOI: 10.1038/s41593-019-0572-3. PMID: 31932767

Sun L, Liu R, Guo F, Wen MQ, Ma XL, Li KY, Sun H, Xu CL, Li YY, Wu MY, Zhu ZG, Li XJ, Yu YQ, Chen Z, Li XY, Duan SM. (2020). Parabrachial nucleus circuit governs neuropathic pain-like behavior. *Nature Communications,* 11. DOI: https://doi.org/10.1038/s41467-020-19767-w, PMID:33239627

Zhou WJ, Jin Y, Meng Q, Zhu X, Bai TJ, Tian YH, Mao Y, Wang LK, Xie W, Zhong N, Luo M, H., Tao WJ, Wang HT, Li J, Qiu BS, Zhou JN, Li XY, Xu H, Wang K, Zhang XC, et al. (2019). A neural circuit for comorbid depressive symptoms in chronic pain. *Nature Neuroscience,* 22:1649-1658. DOI: https://doi.org/10.1038/s41593-019-0468-2, PMID:31451801

[Editors' note: further revisions were suggested prior to acceptance, as described below.]

Reviewer #1 (Recommendations for the authors):In their revised manuscript, the authors have adequately addressed the previous concerns and recommendations raised by this reviewer. Importantly, doing so has improved the overall quality of the manuscript.The manuscript could benefit from additional copy editing to avoid statements such as "PVT neurons innervated by PBN innervation" (Line 39).

We have revised the text. Please see Page 2 Line 40 to 41.